# Generative Modeling with Continuous Flows: Sample Complexity of Flow Matching

## Abstract

Flow matching has recently emerged as a promising alternative to diffusion-based generative models, offering faster sampling and simpler training by learning continuous flows governed by ordinary differential equations. Despite growing empirical success, the theoretical understanding of flow matching remains limited, particularly in terms of sample complexity results. In this work, we provide the first analysis of the sample complexity for flow-matching based generative models without assuming access to the empirical risk minimizer (ERM) of the loss function for estimating the velocity field. Under standard assumptions on the loss function for velocity field estimation and boundedness of the data distribution, we show that a sufficiently expressive neural network can learn a velocity field such that with $\mathcal{O}(\epsilon^{-4})$ samples, such that the Wasserstein-2 distance between the learned and the true distribution is less than $\mathcal{O}(\epsilon)$. The key technical idea is to decompose the velocity field estimation error into neural-network approximation error, statistical error due to the finite sample size, and optimization error due to the finite number of optimization steps for estimating the velocity field. Each of these terms are then handled via techniques that may be of independent interest.

## 1 Introduction

Generative models have recently gained significant attention due to their broad applicability in domains such as image synthesis, computational biology, and reinforcement learning. A common framework among the generative models is diffusion models (Song et al., 2021) that demonstrated the state-of-the-art results in many applications. These models define a generative process by gradually reversing a diffusion process, typically modeled using the framework of Stochastic Differential Equation (SDE). However, training of such models involve learning the score function of the data distribution via score matching, often requiring denoising score estimators and large amounts of compute. Moreover, sampling from diffusion models is computationally intensive, as it typically involves hundreds to thousands of discretization steps, making real-time generation challenging.

Recently, a promising alternative has emerged in the form of flow-matching based generative models (Lipman et al., 2023), which replace the stochastic diffusion process with a deterministic Ordinary Differential Equation (ODE). Instead of estimating scores, these models directly learn a continuous flow that maps noise to data by matching the trajectories of the ODE to an idealized probability flow. This approach circumvents the difficulties associated with score estimation and SDE integration, leading to faster sampling, simpler training objectives, and improved stability. As a result, flow matching provides a compelling and efficient paradigm for generative modeling.

In the diffusion setting, numerous works have analyzed the sample complexity of score matching, establishing conditions under which accurate score estimation is possible (Gupta et al., 2024; Gaur et al., 2025). In particular, the theoretical questions such as: '*how many samples are required for a sufficiently expressive neural network to estimate the score function well enough to generate high-quality samples using a DDPM-type algorithm*' are well studied in the diffusion models.

In contrast, flow matching, despite its growing empirical success (Shifeng et al., 2025), lacks analogous theoretical results. In particular, existing works such as Zhou & Liu (2025) obtain sample complexity results, but do so by assuming access to the ERM of the loss function used to estimate the velocity field, which is an unrealistic assumption in practice. This gap between theory and practice limits our understanding of when and why flow-matching-based models generalize well, and what governs their data efficiency. Addressing this gap is essential for developing a comprehensive theoretical understanding of flow matching.

To address this shortcoming, we provide the first theoretical results on the sample complexity of flow matching without assuming access to the ERM of the loss function used to estimate the velocity field. We ask the following question:

*How many samples are required by a fully expressive neural network to learn a velocity field that enables high-quality sample generation via flow matching, without assuming access to the ERM of the loss function used to estimate the velocity field?*

We answer this question by showing that flow-matching-based methods can achieve a sample complexity of $\mathcal{O}(\epsilon^{-4})$, where $\mathcal{O}(\epsilon)$ is the desired Wasserstein-2 distance between the true and generated distribution. This is achieved by first establishing that the Wasserstein distance between the generated and target distributions can be bounded by a constant multiple of the velocity function estimation error using results from Benton et al. (2024). We then decompose this error into three distinct components, the approximation error that arises due to limited expressiveness of the neural network function class used to approximate the velocity field, the statistical error due to the use of a finite training dataset, and the optimization error, resulting from not reaching the global minimum of the loss function during training due to a finite number of optimization steps. Our analysis achieves a $\mathcal{O}(\epsilon^{-4})$ sample complexity bound, doing so without assuming access to the ERM of the loss function used to estimate the velocity field.

To summarize, the main contributions of our work are as follows:

- **Finite-time sample complexity bounds.** We derive a sample complexity bound of $\mathcal{O}(\epsilon^{-4})$ for flow-matching based generative models.

- **Principled error decomposition.** We introduce a decomposition of the velocity field estimation error into three components: approximation error, statistical error, and optimization error. This framework allows us to isolate and analyze the contribution of each component to the overall sample complexity.

- **First sample complexity bounds for flow matching without assuming access to the ERM of the velocity field estimation loss** This is in contrast to prior works such as Zhou & Liu (2025) where ERM access is assumed.

## 1.1 Related Work

**Theoretical Guarantees and Limitations of Flow Matching.** Recently, Benton et al. (2024) presented error bounds for the generative process using flow matching, assuming that the true velocity field lies within an $\epsilon^2$ neighborhood of the learned velocity field. Their analysis provides error bounds under the assumption of upper bounded $L_2$ approximation error and certain regularity conditions on the data distribution. However, their results do not address sample complexity, a crucial aspect for understanding the number of training samples required for estimating the velocity field, and no such result currently exists.

Zhou & Liu (2025) performed a sample complexity analysis for flow matching models wherein the Wasserstein distance between the true and estimated distributions is bounded by decomposing the error in calculating the velocity function into errors incurred due to the limited approximation power of the neural network function class, as well as the finite sample size. As stated earlier, this work assumes access to the ERM of the velocity estimation loss. Additionally, the sample complexity result obtained in Zhou & Liu (2025) is exponential in the data dimension. Our result avoids this exponential dependence by assuming a constant error due to the limited approximation power of the class of neural networks used to approximate the velocity function. We do this since the focus of our work is to analyze the statistical and optimization errors incurred in flow

matching. Similar assumptions have been used in previous works on diffusion modeling, such as Gupta et al. (2024) and Guan et al. (2025).

**Sample Complexity in Diffusion Models.** Works such as Chen et al. (2023) proved iteration complexity bounds for diffusion models. More recent work, such as Guan et al. (2025), provides convergence analysis for generative modeling in convex domains. Several recent works have analyzed the sample complexity of score-based diffusion models. (Gupta et al., 2024) derives a sample complexity bound assuming access to ERM, and (Gaur et al., 2025) relaxes this assumption, not requiring access to ERM.

## 2 Preliminaries and Problem Formulation

We begin by reviewing the foundations of flow matching models, which aim to learn vector fields that generate smooth transformations between probability distributions over time. Specifically, we set up the formal problem of learning time-dependent velocity functions from finite data samples, under the framework of continuous-time dynamics.

Our approach adopts the ordinary differential equation (ODE) formulation, which serves as a natural setting for describing continuous-time trajectories in space. We conclude this section by precisely formulating our learning objective and articulating the main research question addressed in this work.

A trajectory in this context is a continuous mapping from time $t \in [0, 1]$ to a point in $\mathbb{R}^d$, representing the position of a particle as it evolves over time

$$X : [0, 1] \to \mathbb{R}^d, \quad t \mapsto X_t. \tag{1}$$

The time evolution of this trajectory is governed by a velocity field $u$, which is a time-dependent vector field defined as the mapping

$$u : \mathbb{R}^d \times [0, 1] \to \mathbb{R}^d, \tag{2}$$

so that for each time $t \in [0, 1]$ and $x \in \mathbb{R}^d$, the vector $u(x, t) \in \mathbb{R}^d$ specifies the instantaneous velocity of a particle located at $x$ at time $t$.

Our goal is to find a trajectory $X_t$ that flows consistently with this velocity field, starting from a known initial condition $z$. This leads us to the following ODE

$$\frac{d}{dt} X_t = u(X_t, t), \quad X_0 = z. \tag{3}$$

This formalism underpins many recent advances in generative modeling, including neural ODEs and flow models, where learning an appropriate velocity field is crucial for generating samples that follow a desired probabilistic path (Lipman et al., 2023; Klein et al., 2023). The central challenge we now address is how to learn such a velocity field from empirical data in a principled and data-efficient manner.

**Construction of a generative model via an ODE.** Following flow matching (Lipman et al., 2023), we begin with the construction of a generative model that transforms a simple initial distribution, denoted by $\pi_0$ (e.g., a standard Gaussian), into a more complex target distribution $\pi_1$, which represents the true unknown data distribution. We assume the data distribution $\pi_1$ has an absolutely continuous CDF supported on the set $[0, 1]^d$. A natural and powerful approach to achieve this is by simulating a continuous-time flow using an ordinary differential equation (ODE). This flow is modeled by a time-dependent velocity field through the following ODE:

$$\frac{d}{dt} X_t = u^\theta(X_t, t, X_0), \quad X_0 \sim \pi_0, \tag{4}$$

where $u^\theta : \mathbb{R}^d \times [0, 1] \times \mathbb{R}^d \to \mathbb{R}^d$ is a (time-dependent) vector field parameterized by a neural network with parameters $\theta \in \Theta$. The trajectory $X_t$ evolves over time according to this learned field, with the aim that the induced distribution of $X_1$ closely approximates the target $\pi_1$.

To this end, a general technique (Benton et al., 2024; Lipman et al., 2023) is to minimize the velocity estimation loss function given by

$$\mathcal{L}(\theta) = \int_0^1 \mathbb{E}_{x,z}\left[\left\|u^\theta(x,t,z) - u(x,t,z)\right\|^2\right] dt, \tag{5}$$

where $z \sim \pi_0$, $t \sim \text{Uniform}[0,1]$, and $x \sim (X_t \mid z)$ denotes a sample from the conditional law of $X_t$ given $z$ under the reference probability path. This loss encourages $u^\theta$ to match the target velocity field across the entire time horizon.

**A result from Benton et al. (2024).** The authors in Benton et al. (2024) analyze the error of the flow matching procedure in terms of the TV between the terminal distributions of the true and learned flows. Their analysis is based on a set of assumptions that provide theoretical guarantees for the quality of the learned flow. We restate their main result for completeness.

**Theorem 2.1 (Theorem 1 Benton et al. (2024))** *Suppose that $\pi_0, \pi_1$ are initial and target probability distributions respectively on $\mathbb{R}^d$ , $Y$ is the flow starting in $\pi_0$ with velocity field $u_\theta$, and $\hat{\pi}_1$ is the law of $Y_1$. Also, suppose the following assumptions hold.*

- *(Bound on $L^2$ approximation error). The true and estimated velocity $u_t(x)$ and $u_t^\theta(x)$ satisfy*

$$\mathbb{E}_{x,t,z}\left[\|u^\theta(x,t,z) - u_t(x,z)\|^2\right] \leq \varepsilon^2. \tag{6}$$

  *where $\epsilon$ is a positive real number.*

- *(Existence and uniqueness of smooth flows). For each $x \in \mathbb{R}^d$ and $s \in [0,1]$ there exist unique flows $(Y_{s,t}^x)_{t\in[s,1]}$ and $(Z_{s,t}^x)_{t\in[s,1]}$ starting in $Y_{s,s}^x = x$ and $Z_{s,s}^x = x$ with velocity fields $u^\theta(x,t,z)$ and $u_t(x)$ respectively. Moreover, $Y_{s,t}^x$ and $Z_{s,t}^x$ are continuously differentiable in $x$, $z$, and $t$.*

- *(Regularity of approximate velocity field). The approximate flow $u^\theta(x,t,z)$ is differentiable in all inputs. Also, for each $t \in (0,1)$ there is a constant $L_t$ such that $u^\theta(x,t,z)$ is $L_t$-Lipschitz in $x$.*

*Under the above assumptions, we have*

$$W_2(\hat{\pi}_1, \pi_1) \leq \varepsilon \exp\left\{\int_0^1 L_t \, dt\right\}. \tag{7}$$

**Wasserstein distance and its relation to velocity estimation.** Since $\pi_0, \pi_1$ are the source and target probability distributions on $\mathbb{R}^d$, respectively. For the notational simplicity we use $Y_t$ as the solution of the learned flow governed by $u^\theta$ with initial condition $Y_0 \sim \pi_0$, and $\hat{\pi}_1 = \text{Law}(Y_1)$ be the resulting terminal distribution. Similarly, let $Z_t$ be the solution of the true flow with velocity field $u$, such that $Z_0 \sim \pi_0$ and $\pi_1 = \text{Law}(Z_1)$. Using the standard definition of 2-Wasserstein distance, we have

$$W_2(\hat{\pi}_1, \pi_1) = W_2(\text{Law}(Y_1), \text{Law}(Z_1)) \tag{8}$$

$$\leq \left(\mathbb{E}[\|Y_1 - Z_1\|^2]\right)^{1/2}. \tag{9}$$

To bound the expected squared deviation $\mathbb{E}[\|Y_1 - Z_1\|^2]$, we have the following from Theorem 1 of Benton et al. (2024)

$$\mathbb{E}[\|Y_1 - Z_1\|^2] \leq K^2 \mathbb{E}[\|u^\theta(x,t,z) - u_t(x,z)\|^2] \tag{10}$$

$$\leq K^2 \epsilon^2, \tag{11}$$

where the constant $K$ depends exponentially on the Lipschitz constants of the learned velocity field, and is defined as:

$$K := \exp\left(\int_0^1 L_t \, dt\right). \tag{12}$$

This result provides a theoretical justification for the flow matching framework: if the learned vector field $u^\theta$ closely approximates the true field $u_t$ in an $L_2$ sense, and maintains smoothness and Lipschitz continuity, then the resulting generative distribution $\hat{\pi}_1$ is guaranteed to be close to the target $\pi_1$ in Wasserstein distance.

**Problem statement.** In this work, we investigate the sample complexity required to guarantee that the learned velocity field $u^\theta$, parameterized by a neural network, approximates the true conditional field $u_t(x)$ with small integrated error, i.e.,

$$\mathcal{L}(\theta) = \mathbb{E}_{x,t,z}\left[\|u^\theta(x,t,z) - u_t(x,z)\|^2\right] dt \le \epsilon^2. \tag{13}$$

Note that here $z \sim \pi_0, t \sim U[0,1], x \sim X_t|z,$. Our goal is to understand how the number of training samples influences this error. To this end, we develop a novel theoretical bound by decomposing the total error in the learned velocity field into statistical, approximation, and optimization errors. This decomposition provides insight into the learning dynamics of flow-based generative models. We formalize our framework and main results in the following section.

## 3 Our Approach

In order to bound the total error given in learned velocity field as given in Equation equation 10, we decompose this error into three components. We decompose the loss function in Equation equation 6 as follows.

$$\begin{aligned}
\mathbb{E}\|u^\theta(x,t,z) - u(x,t,z)\|^2 = \mathbb{E}\|u^\theta(x,t,z) - u^{\theta^b}(x,t,z) \\
+ u^{\theta^b}(x,t,z) - u^{\theta^a}(x,t,z) \\
+ u^{\theta^a}(x,t,z) - u(x,t,z)\|^2
\end{aligned} \tag{14}$$

$$\begin{aligned}
\mathbb{E}\left\|u^\theta(x,t,z) - u(x,t,z)\right\|^2 \le 2\,\underbrace{\mathbb{E}\left[\left\|u(x,t,z) - u^{\theta^a}(x,t,z)\right\|^2\right]}_{\mathcal{E}^{\text{approx}}} \\
+ 4\,\underbrace{\mathbb{E}\left[\left\|u^{\theta^a}(x,t,z) - u^{\theta^b}(x,t,z)\right\|^2\right]}_{\mathcal{E}^{\text{stat}}} \\
+ 4\,\underbrace{\mathbb{E}\left\|u^\theta(x,t,z) - u^{\theta^b}(x,t,z)\right\|^2}_{\mathcal{E}^{\text{opt}}},
\end{aligned} \tag{15}$$

We get Equation equation 15 from Equation equation 14 by applying the identity $\|a-b\|^2 \le 2\|a\|^2 + 2\|b\|^2$ twice. The parameters $\theta^a$ and $\theta^b$ are defined as

$$\theta^a = \arg\min_{\theta \in \Theta} \mathbb{E}_{x,t}\left[\|u^\theta(x,t,z) - u(x,t,z)\|^2\right], \tag{16}$$

$$\theta^b = \arg\min_{\theta \in \Theta} \frac{1}{n}\sum_{i=1}^n \left\|u^\theta(x_i,t_i,z_i) - u_t(x_i)\right\|^2, \tag{17}$$

and we denote $u^{\theta^a}$ and $u^{\theta^b}$ as the neural networks associated with the parameters $\theta^a$ and $\theta^b$, respectively.

In the above the parameters $\theta^a$ defines the ideal or population-level parameters. It is the value of $\theta$ that minimizes the expected squared error between the model velocity field $u^\theta(x,t,z)$ and the true velocity field $u(x,t,z)$, averaged over the entire data distribution. Moreover, $\theta^b$ defines the empirical minimzer. It is the value of $\theta$ that minimizes the empirical average of the squared error between the model and true velocity fields, based on a finite set of $n$ samples $\{x_i, t_i, z_i\}_{i=1}^n$.

In the above, the *approximation error* $\mathcal{E}_t^{\text{approx}}$ captures the error due to the limited expressiveness of the velocity field $\{u^\theta\}_{\theta \in \Theta}$. The *statistical error* $\mathcal{E}_t^{\text{stat}}$ is the error from using a finite sample size. Finally, the *optimization error* $\mathcal{E}_t^{\text{opt}}$ is due to not reaching the global minimum during training.

To formally prove the sample complexity results, we make the following assumptions that are commonly used in previous works on the sample complexity of diffusion models (Block et al., 2020; Gupta et al., 2024).

**Assumption 3.1 ((PL) condition.)** *The loss $\mathcal{L}(\theta)$ for all $t \in [0,1]$ satisfies the Polyak–Łojasiewicz condition, i.e., there exists a constant $\mu > 0$ such that*

$$\frac{1}{2}\|\nabla\mathcal{L}(\theta)\|^2 \geq \mu\left(\mathcal{L}(\theta) - \mathcal{L}(\theta^*)\right), \quad \forall\, \theta \in \Theta, \tag{18}$$

*where $\theta^* = \arg\min_{\theta\in\Theta}\mathcal{L}(\theta)$ denotes the global minimizer of the population loss.*

The Polyak-Łojasiewicz (PL) condition is much weaker than strong convexity and often holds in non-convex settings, including overparameterized neural networks trained with mean squared error (Liu et al., 2022; Gaur et al., 2025). Prior work on diffusion model sample complexity (Gupta et al., 2024; Block et al., 2020) as well as flow matching (Zhou & Liu, 2025) assumes access to an exact empirical risk minimizer (ERM) for the score estimation loss. However, this assumption is limiting in practice, as exact ERM solutions are rarely attainable.

**Assumption 3.2 (Smoothness and stochastic gradient oracle)** *The population loss $\mathcal{L}(\theta)$ is $\alpha$-smooth in $\theta$, i.e., for all $\theta, \theta' \in \Theta$,*

$$\|\nabla_\theta\mathcal{L}(\theta) - \nabla_\theta\mathcal{L}(\theta')\| \leq \alpha\,\|\theta - \theta'\|. \tag{19}$$

*Let $\{\mathcal{F}_i\}_{i\geq 0}$ be the filtration generated by the algorithm up to iteration $i$ (so $\theta_i$ is $\mathcal{F}_i$-measurable). At iteration $i$, we draw a fresh random seed $\xi_i = \{x_i, t_i, z_i\}$ (e.g., data/latent variables used to form the stochastic loss at time $t$), independent of $\mathcal{F}_i$, and define the stochastic gradient*

$$g(\theta_i, \xi_i) \;:=\; \nabla_\theta\widehat{\mathcal{L}}(\theta_i; \xi_i), \tag{20}$$

*(Conditional unbiasedness).* *For all $i$,*

$$\mathbb{E}[g(\theta_i, \xi_i) \mid \mathcal{F}_i] \;=\; \nabla_\theta\mathcal{L}(\theta_i). \tag{21}$$

*Equivalently, for any $(\theta, \xi_i)$, if $g(\theta, \xi) := \nabla_\theta\widehat{\mathcal{L}}(\theta; \xi)$ with $\xi$ fresh,*

$$\mathbb{E}_\xi[g(\theta, \xi)] \;=\; \nabla_\theta\mathcal{L}(\theta). \tag{22}$$

*(Conditional bounded variance).* *There exists $\sigma^2 > 0$ such that for all $i$,*

$$\mathbb{E}\left[\|g(\theta_i, \xi_i) - \nabla_\theta\mathcal{L}(\theta_i)\|^2 \,\Big|\, \mathcal{F}_i\right] \leq \sigma^2. \tag{23}$$

The bounded gradient variance assumption is standard in SGD analysis (Koloskova et al., 2022; Ajalloeian & Stich, 2020) and holds under mild conditions such as Lipschitz activations (e.g., GELU), bounded inputs, and standard initialization (Allen-Zhu et al., 2019; Du et al., 2019). Smoothness further stabilizes gradients, improving convergence for methods like SGD and Adam.

**Remark (online vs. finite-dataset multi-epoch training).** Assumption 3.2 models the *online / streaming* setting in which each iteration uses a fresh i.i.d. draw $\xi_i$ independent of the past. This corresponds to one-pass SGD / stochastic approximation for the population objective $L(\theta)$. Analyzing multi-epoch SGD on a fixed finite dataset (with data reuse across epochs) requires additional arguments to control the dependence introduced by repeated sampling and is outside the scope of the present work.

**Assumption 3.3 (Approximation error)** *There exists a neural network parameter $\theta \in \Theta$ such that*

$$\mathbb{E}_{x,t,z}\|u_\theta(x,t,z) - u_t(x,z)\|^2 \leq \epsilon_{approx} \tag{24}$$

The approximation error assumptiom describes the error due to neural network parametrization. In learning theory, it is common to treat the *approximation error* of a class of functions as a constant so that analysis

can focus on the estimation/ optimization terms dependent on the sample. In PAC-Bayesian analyses, approximation errors are denoted by a constant once the class is fixed (Mai, 2025). In (NTK/RKHS) analyses of neural networks, where it is assumed the target function lies in, or is well approximated by the specified function class, the misspecification error is represented as a constant term (Bing et al., 2025). Note that diffusion model analyses such as Gupta et al. (2024) and Guan et al. (2025) also make the same assumptions.

**Assumption 3.4 (Smoothness of velocity field)** *For each $x \in \mathbb{R}^d$ and $s \in [0,1]$, there exist unique flows $(Y_{s,t}^x)_{t \in [s,1]}$ and $(Z_{s,t}^x)_{t \in [s,1]}$ satisfying $Y_{s,s}^x = Z_{s,s}^x = x$, with respective velocity fields $v_\theta(x,t)$ and $v^X(x,t)$. Moreover, $Y_{s,t}^x$ and $Z_{s,t}^x$ are continuously differentiable in $x$, $s$, and $t$. Additionally, the velocity field $u^\theta(x,t,z)$ is Lipschitz in $\theta$, that is, there exists a constant $L > 0$ such that for all $\theta_1, \theta_2 \in \Theta$ and $x, t, z$ we have*

$$\|u^{\theta_1}(x,t,z) - u^{\theta_2}(x,t,z)\| \leq L\|\theta_1 - \theta_2\|. \tag{25}$$

The assumption regarding the smoothness of the velocity field ensures the well-posedness of the associated flow dynamics governed by $u_t^\theta(x)$. Specifically, Lipschitz continuity in $x$ guarantees the existence and uniqueness of solutions to the corresponding ordinary differential equation (ODE). Moreover, it implies stability of the flow with respect to perturbations in the initial condition, which is crucial for both theoretical analysis and practical implementations involving generative modeling.

**Gaussian Probability Path.** We use a Gaussian probability path as the time-indexed marginal distribution $\mu_t$ via a Gaussian *conditional* path. For a given latent variable $z \sim \pi_0$ sampled from the initial distribution, we define

$$X_t \mid z \sim \mathcal{N}(tz, (1-t)^2 I_d). \tag{26}$$

This choice yields a smooth family $\{\mu_t\}_{t \in [0,1]}$ of analytically tractable intermediate marginals that is convenient for both training and theoretical analysis.

This construction follows the idea of probability flows, where intermediate distributions are designed to be simple and closed-form. Similar to the linear conditional flow in Liu et al. (2023), the Gaussian path allows closed-form expressions for the velocity field and supports stable training without stochastic score matching. Hence, the Gaussian path serves as a constructive prior, a design choice that promotes smoothness, clarity, and differentiability in the probability flow.

Furthermore, under this construction, the dynamics of the probability flow can be fully characterized by the evolution of the mean and covariance enabling efficient computation and theoretical analysis. The associated velocity field $u(x,t,z)$ has the form (Lipman et al., 2023)

$$u(x,t,z) = \frac{z-x}{1-t}. \tag{27}$$

*Validity of the velocity field.* Indeed, for each fixed $z$ the ODE $\dot{x}_t = u_t(x_t, z)$ admits the explicit solution

$$x_t = (1-t)x_0 + tz. \tag{28}$$

If $x_0 \sim \mathcal{N}(0, I_d)$ is independent of $z$, then $x_t \mid z \sim \mathcal{N}(tz, (1-t)^2 I_d)$, matching the above conditional Gaussian path. Thus $u_t(x,z)$ is the (probability-flow) velocity field that generates the stated conditional marginals for all $t \in [0,1)$.

Smoothness and bounded gradient variance both implied by sub-Gaussianity are mild assumptions and typically hold for standard networks with ReLU or GELU activations trained on well-behaved data. These assumptions are widely used in recent studies on the optimization landscape of generative and diffusion models (Salimans & Ho, 2022; Liu et al., 2022).

## 4  Theoretical Results

This section presents a comprehensive sample complexity result for flow matching.

**Theorem 4.1** *Let assumptions 3.1–3.4 hold. Suppose that the velocity field $u_t^\theta(x,t,z)$ is parameterized by a neural network with width $W$ and depth $D$ and $\epsilon$ is a positive real number.. Then for any confidence level $\delta \in (0,1)$, if the number of i.i.d. training samples $n$ satisfies*

$$n = \Omega\left(\frac{(W)^{2D-2}d^2}{\epsilon^4}\log\frac{2}{\delta}\right), \tag{29}$$

*and the learning rate for the $i$th* online (one-pass) SGD *step (one fresh sample per step), satisfies $\eta_i = \frac{\alpha}{i+\gamma}$ where $\alpha \cdot \mu > 1$ and $\gamma > \alpha \cdot \kappa$. Then with probability at least $1 - 4\delta$, the learned velocity field satisfies*

$$\mathbb{E}_{x,t,z}\left\|u^\theta(x,t,z) - u(x,t,z)\right\|^2 \le \epsilon^2 + \epsilon_{approx}, \tag{30}$$

*and the Wasserstein distance between the learned and true distribution satisfies*

$$W_2(\hat{\pi}_1, \pi_1) \le \mathcal{O}(\epsilon) + \epsilon_{approx}. \tag{31}$$

The proof of this theorem relies on bounding each of the errors given in Equation equation 15. Below, we bound each of these in the following lemmas.

**Lemma 4.1 (Approximation Error)** *Let $W$ and $D$ denote the width and depth of the neural network architecture, respectively, and let $d$ represent the input data dimension. Then, under Assumption 3.3, we have*

$$\mathcal{E}^{\mathrm{approx}} \le \epsilon_{approx} \tag{32}$$

This result directly follows from Assumption 3.3 and the definition of $\mathcal{E}^{\mathrm{approx}}$.

**Lemma 4.2 (Statistical Error)** *Let $n$ denote the number of samples used to estimate the velocity field and $\epsilon$ a positive real number. Then, under assumptions 3.1 and 3.4 with probability at least $1 - 2\delta$, we have*

$$\mathcal{E}^{\mathrm{stat}} \le \mathcal{O}\left((W)^{D-1}d \cdot \sqrt{\frac{\log\frac{2}{\delta}}{n}}\right). \tag{33}$$

**Proof Outline** We present the outline of the proof, full details are deferred to the Appendix.

Recall the population loss at time $t \in [0,1]$ is

$$\mathcal{L}(\theta) = \mathbb{E}_{x,t,z}\left\|u^\theta(x,t,z) - u_t(x)\right\|^2, \tag{34}$$

The corresponding empirical loss over $n$ samples $\{z_i, t_i, x_i,\}_{i=1}^n$ is given by

$$\widehat{\mathcal{L}}(\theta) = \frac{1}{n}\sum_{i=1}^n\left\|u^\theta(x_i,t_i,z_i) - u(x_i,t_i,z_i)\right\|^2. \tag{35}$$

Here $x_i$ denotes the $i^{th}$ data point where $z_i$ sampled from $\pi_0$, $t_i$ is sampled from $U[0,1]$ and $x_i$ is sampled from $x_i \sim X_{t_i}|z_i$. From equations equation 16 and equation 17 we have that $\theta^a$ and $\theta^b$ are the minimizers of $\mathcal{L}(\theta)$ and $\widehat{\mathcal{L}}(\theta)$, respectively. We denote the corresponding velocity fields as $u^a := u^{\theta^a}$ and $u^b := u^{\theta^b}$ respectively. Thus, we have

$$\mathcal{L}(\theta^b) - \mathcal{L}(\theta^a) \le \mathcal{L}(\theta^b) - \mathcal{L}(\theta^a) + \widehat{\mathcal{L}}(\theta^a) - \widehat{\mathcal{L}}(\theta^b), \tag{36}$$

We get the inequality by adding the term $\widehat{\mathcal{L}}(\theta^a) - \widehat{\mathcal{L}}(\theta^b)$ to the right hand side of Equation equation 36, this is a positive quantity since $\theta^b$ is the minimizer of $\hat{\mathcal{L}}(\theta)$, where the added term is non-negative due to the empirical optimality of $\theta^b$.

$$\left|\mathcal{L}(\theta^b) - \mathcal{L}(\theta^a)\right| \leq \underbrace{\left|\mathcal{L}(\theta^b) - \widehat{\mathcal{L}}(\theta^b)\right|}_{(I)} + \underbrace{\left|\mathcal{L}(\theta^a) - \widehat{\mathcal{L}}(\theta^a)\right|}_{(II)}. \tag{37}$$

We get the inequality in Equation equation 37 by taking the absolute value on both sides of the Equation equation 36 and applying the triangle inequality on the right-hand side of the resulting Equation. We now bound each of the terms I and II separately as follows. To bound (I) and (II),we need to apply results which require the loss function $\mathcal{L}(\theta)$. However, note that $x$ is unbounded, which in turn implies that $\mathcal{L}(\theta)$ is unbounded. Thus,we first define truncated versions of the velocity fields as

$$(v(x,t,z))_k = \begin{cases} (u(x,t,z))_k & \text{if } \left|\left(\frac{x-tz}{1-t}\right)_k\right| \leq \kappa, \\ 0 & \text{otherwise,} \end{cases}$$

$$\left(v^\theta(x,t,z)\right)_k = \begin{cases} \left(u^\theta(x,t,z)\right)_k & \text{if } \left|\left(\frac{x-tz}{1-t}\right)_k\right| \leq \kappa, \\ 0 & \text{otherwise.} \end{cases}$$

Here $(v(x,t,z))_k$, $\left(v^\theta(x,t,z)\right)_k$ and $\left(\frac{x-tz}{1-t}\right)_k$ represent the $k^{th}$ co-ordinates of $v(x,t,z)$, $v^\theta(x,t,z)$ and $\left(\frac{x-tz}{1-t}\right)$ respectively where $k \in \{1,\cdots,d\}$.

We define the corresponding loss functions as

$$\mathcal{L}'(\theta) = \mathbb{E}_{x,t,z}\left\|v^\theta(x,t,z) - v(x,t,z)\right\|^2, \tag{38}$$

$$\widehat{\mathcal{L}}'(\theta) = \frac{1}{n}\sum_{i=1}^n\left\|v^\theta(x_i,t_i,z_i) - v(x_i,t_i,z_i)\right\|^2. \tag{39}$$

Note that the bounded velocity filed is defined as $v(x,t.z) = \frac{z-x}{1-t}$, this can be decomposed as

$$v(x,t.z) = \left|\frac{z-x}{1-t}\right| \leq \left|\frac{x-tz}{1-t}\right| + |z| \tag{40}$$

Since $\left|\frac{x-tz}{1-t}\right|$ is upper bounded by construction and $z$ is defined over a bounded set, this implies that the input $\frac{z-x}{1-t}$ to the loss function $\mathcal{L}'(\theta)$ and $\hat{\mathcal{L}}'(\theta)$ is bounded. Thus, for a fixed value of $\theta$ both $\mathcal{L}'(\theta)$ and $\hat{\mathcal{L}}'(\theta)$ are bounded. Now consider the function class $\Theta'' := \{\theta^a, \theta^b\}$. For a bounded input $v(x,t.z)$, $\mathcal{L}'(\theta)$ and $\hat{\mathcal{L}}'(\theta)$ are bounded for all $\theta \in \Theta''$. Therfore from Lemma D.1 we have with probability at least $1 - \delta$

$$\left|\mathcal{L}'(\theta) - \widehat{\mathcal{L}}'(\theta)\right| \leq \mathcal{O}\left((W)^{D-1}d \cdot \sqrt{\frac{\log\frac{1}{\delta}}{n}}\right),$$

$$\forall \, \theta \in \Theta''. \tag{41}$$

Applying this to the terms (I) and (II), equation equation 37 now becomes

$$\left|\mathcal{L}(\theta^b) - \mathcal{L}(\theta^a)\right| \leq \mathcal{O}\left((W)^{D-1}d \cdot \sqrt{\frac{\log\frac{1}{\delta}}{n}}\right). \tag{42}$$

Using the quadratic growth property of Polyak–Łojasiewicz (PL) functions (Karimi et al., 2020), we have

$$\|\theta^a - \theta^b\|^2 \leq \mu \cdot \left|\mathcal{L}(\theta^a) - \mathcal{L}(\theta^b)\right|, \tag{43}$$

where $\mu$ is the PL constant. Also, Lipschitz continuity of the velocity fields in $\theta$ leads to

$$\|v^{\theta^a}(x,t,z) - v^{\theta^b}(x,t,z)\|^2$$

$$\leq L \cdot \|\theta^a - \theta^b\|^2 \tag{44}$$

$$\leq L \cdot \mu \cdot |\mathcal{L}(\theta^a) - \mathcal{L}(\theta^b)| \tag{45}$$

$$\leq \mathcal{O}\left((W)^{D-1}d \cdot \sqrt{\frac{\log \frac{1}{\delta}}{n}}\right). \tag{46}$$

Since $v^{\theta^a}(x,t) = u^{\theta^a}(x,t)$ and $v^{\theta^b}(x,t) = u^{\theta^b}(x,t)$ for all $x$ in the truncated domain (which dominates the mass under $\mu_t$), by taking expectation over $x, t, z$ we finally obtain

$$\mathbb{E}_{x,t,z}\left\|u^{\theta^a}(x,t,z) - u^{\theta^b}(x,t,z)\right\|^2$$

$$\leq \mathcal{O}\left((W)^{D-1}d \cdot \sqrt{\frac{\log \frac{1}{\delta}}{n}}\right). \tag{47}$$

This completes the proof. For a more detailed version of this proof see Appendix 4.2.

Next we bound the optimization error in the following lemma. The detailed proof is deferred to the Appendix A.

**Lemma 4.3 (Optimization Error)** *Let $n$ be the number of* online (one-pass) *stochastic-gradient oracle calls used to estimate the velocity field and $\epsilon$ a positive real number. If the learning rate for the ith online step (one fresh sample per step) satisfies $\eta_i = \frac{\alpha}{i+\gamma}$ where $\alpha \cdot \mu > 1$ and $\gamma > \alpha \cdot \kappa$, then under Assumptions 3.1, 3.2, and 3.4, the optimization error due to running only $n$ steps satisfies with probability at least $1 - 2\delta$*

$$\mathcal{E}^{\mathrm{opt}} \leq \mathcal{O}\left((W)^{D-1}d \cdot \sqrt{\frac{\log \frac{1}{\delta}}{n}}\right). \tag{48}$$

**Proof Outline**

By the smoothness of $\mathcal{L}(\theta)$ (Assumption 3.2), we get

$$\mathcal{L}(\theta_{i+1}) \leq \mathcal{L}(\theta_i) + \langle \nabla \mathcal{L}(\theta_i), \theta_{i+1} - \theta_i \rangle$$

$$+ \frac{\kappa}{2}\|\theta_{i+1} - \theta_i\|^2. \tag{49}$$

Taking expectation and using unbiasedness of the stochastic gradient, along with bounded variance and gradient norm assumptions (Assumption 3.3), we get

$$[\mathcal{L}(\theta_{i+1})] \leq \mathcal{L}(\theta_i) - \left(\eta_i - \frac{\kappa\eta_i^2}{2}\right)\|\nabla \mathcal{L}(\theta_i)\|^2$$

$$+ \frac{\kappa\eta_i^2\sigma^2}{2}. \tag{50}$$

From the Polyak-Łojasiewicz (PL) property and the fact that $0 < \eta_i < \frac{1}{\kappa}$, we then obtain

$$[\mathcal{L}(\theta_{i+1}) - \mathcal{L}^*] \leq (1 - \mu\eta_i)[\mathcal{L}(\theta_i) - \mathcal{L}^*] + \frac{\kappa\eta_i^2\sigma^2}{2}. \tag{51}$$

Further, in Appendix B, we show that with the chosen step sizes $\eta_k$, we obtain the following

$$[\mathcal{L}(\theta_n) - \mathcal{L}^*] \leq \mathcal{O}\left(\frac{1}{n}\right). \tag{52}$$

Noting that we use one fresh sample per iteration (online / one-pass setting), the number of stochastic gradient oracle calls equals $n$, and the optimization suboptimality decays as $\mathcal{O}(1/n)$ in $n$. Next, using the Lipschitz continuity of the velocity field and the quadratic growth property of PL functions Karimi et al. (2020) we get

$$
\mathbb{E}\|u^\theta(x,t,z) - u^{\theta^a}(x,t,z)\|^2 \le \mu \, |\mathcal{L}(\theta_n) - \mathcal{L}^*|
$$
$$
\le \mathcal{O}\left(\frac{1}{n}\right). \tag{53}
$$

Also note that $u^\theta$ corresponds to the parameter $\theta_n$ since $\theta_n$ our estimate of $\theta$ obtained from SGD. Now, using the triangle inequality, we have that

$$
\mathbb{E}\|u^\theta(x,t,z) - u^{\theta^b}(x,t,z)\|^2
$$
$$
\le 2\mathbb{E}\|u^\theta(x,t,z) - u^{\theta^a}(x,t,z)\|^2
$$
$$
+ 2\mathbb{E}\|u^{\theta^b}(x,t,z) - u^{\theta^a}(x,t,z)\|^2 \tag{54}
$$

Now from equation 53 and equation 47, we obtain with probability at least $1 - \delta$

$$
\mathbb{E}\|u^\theta(x,t,z) - u^{\theta^b}(x,t,z)\|^2
$$
$$
\le \mathcal{O}\left((W)^{D-1}d \cdot \sqrt{\frac{\log\frac{1}{\delta}}{n}}\right). \tag{55}
$$

The details of the proof are given in Appendix B.

Now, to complete the proof of our main result (Theorem 4.1), we combine Lemmas 4.1–4.3 to obtain

$$
\mathbb{E}\left[\left\|u^\theta(x,t,z) - u(x,t,z)\right\|^2\right]
$$
$$
\le \epsilon_{approx}
$$
$$
+ \mathcal{O}\left((W)^{D-1}d\sqrt{\frac{\log(2/\delta)}{n}}\right). \tag{56}
$$

Choosing $n = \mathcal{O}\left(\frac{(W)^{2D-2}d^2}{\epsilon^4}\log(2/\delta)\right)$ gives

$$
\mathbb{E}_{x,t,z}\left[\left\|u^\theta(x,t,z) - u_t(x)\right\|^2\right] \le \epsilon^2 + \epsilon_{approx}. \tag{57}
$$

The bound on the Wasserstein distance between the true and the learned distribution follows from Equations equation 8, equation 10 and equation 12, i.e.,

$$
W_2(\hat{\pi}_1, \pi_1) \le \mathcal{O}(\epsilon) + \epsilon_{approx}. \tag{58}
$$

This completes the proof.

## 5 Conclusion

In this paper, we analyze the sample complexity of training flow matching models via neural network-based velocity estimation. We establish a sample complexity bound of notably avoiding exponential dependence on the data dimension. To the best of our knowledge, this is the first formal sample complexity result for flow matching methods, and uniquely, it is derived under the realistic setting where exact empirical risk minimization is not assumed.

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

## Appendix

In this appendix, we provide the missing proofs and additional theoretical results referenced in the main paper. We also include the flow matching algorithm used for training the generative model.

## A  Proof of Lemma 4.2

**Proof A.1** *Let us define the population loss at time $t$ for $t \in [0,1]$ as*

$$\mathcal{L}(\theta) = \mathbb{E}_{x,t,z} \left\| u^\theta(x,t,z) - u_t(x) \right\|^2, \tag{59}$$

*where $u_t^\theta$ denotes the velocity estimated by a neural network parameterized by $\theta \in \Theta$.*

*The corresponding empirical loss is defined as*

$$\widehat{\mathcal{L}}(\theta) = \frac{1}{n} \sum_{i=1}^{n} \left\| u^\theta(x_i, z_i, t_i) - u_{t_i}(x_i) \right\|^2. \tag{60}$$

*Here $x_i$ denotes the $i^{th}$ data point where $z_i$ sampled from $\pi_0$, $t_i$ is sampled from $U[0,1]$ and $x_i$ is sampled from $x \sim X_{t_i}(.|z_i)$. Since $\theta^a$ and $\theta^b$ are the minimizers of $\mathcal{L}(\theta)$ and $\widehat{\mathcal{L}}'(\theta)$, respectively, with corresponding velocity fields $u^a$ and $u^b$. By the definitions of minimizers, we can write*

$$\mathcal{L}(\theta^b) - \mathcal{L}(\theta^a) \leq \mathcal{L}(\theta^b) - \mathcal{L}(\theta^a) + \widehat{\mathcal{L}}(\theta^a) - \widehat{\mathcal{L}}(\theta^b) \tag{61}$$

*Note that the right-hand side of the above equation is less than the left-hand side since we have added the quantity $\widehat{\mathcal{L}}'(\theta^a) - \widehat{\mathcal{L}}'(\theta^b)$ which is strictly positive as $\theta^b$ is the minimizer of the function $\widehat{\mathcal{L}}'(\theta)$ by definition. We then take the absolute value on both sides of the equation to get*

$$|\mathcal{L}(\theta^b) - \mathcal{L}(\theta^a)| \leq \underbrace{\left| \mathcal{L}(\theta^b) - \widehat{\mathcal{L}}(\theta^b) \right|}_{(I)} + \underbrace{\left| \mathcal{L}(\theta^a) - \widehat{\mathcal{L}}(\theta^a) \right|}_{(II)}. \tag{62}$$

*We now bound terms (I) and (II) using generalization results. From Lemma D.1 (Theorem 26.5 of Shalev-Shwartz & Ben-David (2014)), if the loss function $\widehat{\mathcal{L}}(\theta)$ is uniformly bounded over the parameter space $\Theta'' = \{\theta^a, \theta^b\}$, then with probability at least $1 - \delta$, we have*

$$\left| \mathcal{L}(\theta) - \widehat{\mathcal{L}}'(\theta) \right| \leq \widehat{R}(\Theta'') + \mathcal{O}\left( \sqrt{\frac{\log \frac{1}{\delta}}{n}} \right), \quad \forall \, \theta \in \Theta'' \tag{63}$$

*where $\widehat{R}(\Theta'')$ denotes the empirical Rademacher complexity of the function class restricted to $\Theta''$.*

*Now since $x$ is not bounded, this result does not hold. We then define the following two functions*

$$(v_t(x))_k = \begin{cases} (u_t(x|z))_k, & if \left| \frac{x-tz}{1-t} \right|_k \leq \kappa, \\ 0, & otherwise, \end{cases}$$

$$\left( v^\theta(x,t) \right)_k = \begin{cases} \left( u^\theta(x,t,z) \right)_k, & if \left| \frac{x-tz}{1-t} \right|_k \leq \kappa, \\ 0, & otherwise. \end{cases}$$

*Here $(v_t(x|z))_k$ and $\left( v^\theta(x,t) \right)_k$ represent the $k^{th}$ co-ordinates of $v(x,t)$ and $v^\theta(x,t)$, respectively where $k \in \{1, \cdots, d\}$.*

*Using above we have*

$$\mathcal{L}'(\theta) = \mathbb{E}_{x,t,z} \left\| v^\theta(x,t,z) - v_t(x) \right\|^2, \tag{64}$$

$$\widehat{\mathcal{L}}'(\theta) = \frac{1}{n} \sum_{i=1}^{n} \left\| v^\theta(x_i, t_i, z_i) - v_{t_i}(x_i) \right\|^2. \tag{65}$$

Here $(v_t(x))_k$, $(u_t(x))_k$, $\left(v^\theta(x,t,z)\right)_k$ and $\left(u^\theta(x,t,z)\right)_k$ denote the $k^{th}$ co-ordinate of $v_t(x)$, $u_t(x)$, $v^\theta(x,t,z)$ and $u^\theta(x,t,z)$ respectively, where we have $k \in \{1, \cdots, d\}$.

Note that the functions $v_t(x)$ and $v_t^\theta(x)$ are uniformly bounded.

Now using Lemma D.1 we have with probability at least $1 - \delta$,

$$\left| \mathcal{L}'(\theta) - \widehat{\mathcal{L}}'(\theta) \right| \leq \widehat{R}(\theta) + \mathcal{O}\left( \sqrt{\frac{\log \frac{1}{\delta}}{n}} \right), \qquad \forall\, \theta \in \Theta''. \tag{66}$$

Since $\Theta'' = \{\theta_a, \theta_b\}$ is a finite class (just two functions). We apply Lemma D.2 to bound the empirical Rademacher complexity $\widehat{R}(\theta)$ and thus, with probability at least $1 - \delta$ we have that

$$\left| \mathcal{L}'(\theta) - \widehat{\mathcal{L}}'(\theta) \right| \leq \mathcal{O}\left( \frac{(W)^{D-1} d\kappa}{n} \right) + \mathcal{O}\left( \sqrt{\frac{\log \frac{1}{\delta}}{n}} \right), \qquad \forall\, \theta \in \Theta''. \tag{67}$$

This yields that with probability at least $1 - \delta$ we have

$$\left| \mathcal{L}'(\theta) - \widehat{\mathcal{L}}'(\theta) \right| \leq \mathcal{O}\left( (W)^{D-1} d\kappa \cdot \sqrt{\frac{\log \frac{1}{\delta}}{n}} \right), \qquad \forall\, \theta \in \Theta'' \tag{68}$$

Now consider the probability of the event

$$A_{i,k} = \left\{ \left| \left( \frac{(x_i) - t_i(z_i)}{1 - t} \right)_k \right| \geq \kappa \right\} \tag{69}$$

We have the probability of this event to be upper-bounded as

$$P\left( \left| \frac{(x_i) - t_i(z_i)}{1 - t} \right|_k \geq \kappa \right) = \mathbb{E}_{z_i, t_i}\left( P\left( \left| \left( \frac{(x_i) - t_i(z_i)}{1 - t} \right)_k \right| \geq \kappa \middle| z_i, t_i \right) \right) \tag{70}$$

$$\leq \mathbb{E}_{z_i, t_i}\left( \exp\left( -\frac{\kappa^2}{2C} \right) \right) \tag{71}$$

$$\leq \exp\left( -\frac{\kappa^2}{2C} \right) \tag{72}$$

Setting $\kappa = \sqrt{2C \cdot \log\left( \frac{dn}{\delta} \right)}$, we have

$$P\left( \left| \left( \frac{(x)_i - t_i(z)_i}{1 - t} \right)_k \right| \geq \kappa \right) \leq \frac{\delta}{dn} \tag{73}$$

If we denote the event $A = \cup_{i,k} A_{i,k}$, then by union bound we have $P(A) = P(\cup_{i,k} A_{i,k}) \leq \sum_{i,k} P(A_{i,k}) \leq \delta$.

Let event $B$ denote the failure of the generalization bound, i.e.,

$$B := \left\{ \left| \mathcal{L}'(\theta) - \widehat{\mathcal{L}}'(\theta) \right| > \widehat{R}(\Theta'') + \mathcal{O}\left( \sqrt{\frac{\log \frac{1}{\delta}}{n}} \right) \right\}. \tag{74}$$

From above, we know $\mathbb{P}(B) \leq \delta$ under the boundedness condition. Therefore, by the union bound, we have

$$\mathbb{P}(A \cup B) \leq \mathbb{P}(A) + \mathbb{P}(B) \leq 2\delta, \tag{75}$$
$$\implies \mathbb{P}(A^c \cap B^c) = 1 - P(A \cup B) \geq 1 - 2\delta. \tag{76}$$

On this event $(A^c \cap B^c)$, we have $\widehat{\mathcal{L}}'(\theta) = \widehat{\mathcal{L}}(\theta)$.

Now consider the following,

$$|\mathcal{L}(\theta^b) - \mathcal{L}(\theta^a)| \leq \left| \mathcal{L}(\theta^b) - \widehat{\mathcal{L}}'_t(\theta^b) \right| + \left| \mathcal{L}(\theta^a) - \widehat{\mathcal{L}}'_t(\theta^a) \right| \tag{77}$$

$$= \left| \mathcal{L}(\theta^b) - \widehat{\mathcal{L}}'_t(\theta^b) + \mathcal{L}'(\theta^b) - \mathcal{L}'(\theta^b) \right|$$
$$+ \left| \mathcal{L}(\theta^a) - \widehat{\mathcal{L}}'_t(\theta^a) + \mathcal{L}'(\theta^a) - \mathcal{L}'(\theta^a) \right| \tag{78}$$

$$\leq \left| \mathcal{L}'(\theta^b) - \widehat{\mathcal{L}}'_t(\theta^b) \right| + |\mathcal{L}(\theta^b) - \mathcal{L}'(\theta^b)|$$
$$+ \left| \mathcal{L}'(\theta^a) - \widehat{\mathcal{L}}'_t(\theta^a) \right| + |\mathcal{L}(\theta^a) - \mathcal{L}'(\theta^a)| \tag{79}$$

The first equation 77 is the same as equation 62 with $\widehat{\mathcal{L}}(\theta)$ replaces by $\widehat{\mathcal{L}}'(\theta)$. We obtain equation 78 from equation 77 by adding the terms $\widehat{\mathcal{L}}'_t(\theta^a) + \mathcal{L}'(\theta^a)$ and $\widehat{\mathcal{L}}'_t(\theta^b) + \mathcal{L}'(\theta^b)$ to the two terms on the right-hand side of Equation equation 77. Equation equation 79 follows from Equation equation 78 by applying the triangle inequality to both the term on the right-hand side of Equation equation 78.

Note that the terms $\left| \mathcal{L}'(\theta^a) - \widehat{\mathcal{L}}'_t(\theta^a) \right|$ and $\left| \mathcal{L}'(\theta^b) - \widehat{\mathcal{L}}'_t(\theta^b) \right|$ can be upper bounded using the result in Equation equation 68 to get the following with probability at least $1 - 2.\delta$

$$|\mathcal{L}(\theta^b) - \mathcal{L}(\theta^a)| \leq \mathcal{O}\left( (W)^{D-1} d\kappa \cdot \sqrt{\frac{\log \frac{1}{\delta}}{n}} \right) + |\mathcal{L}(\theta^b) - \mathcal{L}'(\theta^b)| + |\mathcal{L}(\theta^a) - \mathcal{L}'(\theta^a)| \tag{80}$$

In order to bound terms of the form $|\mathcal{L}(\theta) - \mathcal{L}'(\theta)|$ we have the following

$$|\mathcal{L}(\theta) - \mathcal{L}'(\theta)| = \sum_{k=1}^{d} \mathbb{E}_{x_k,t,z_k} |(u^\theta(x,t,z))_k - (u_t(x))_k|^2 - \mathbb{E}_{x_k,t,z_k} |(v^\theta(x,t,z))_k - (v_t(x))_k|^2 \tag{81}$$

$$= \sum_{k=1}^{d} \mathbb{E}_{x_k,t,z_k} \left( |(u^\theta(x,t,z))_k - (u_t(x))_k|^2 \mathbf{1}_{\left|\left(\frac{x-tz}{1-t}\right)_k\right| \geq \kappa} \right) \tag{82}$$

$$= \sum_{k=1}^{d} \mathbb{E}_{x_k,t,z_k} \left( \left| \left(\frac{z-x}{1-t}\right)_k - (u^\theta(x,t,z))_k \right|^2 \mathbf{1}_{\left|\left(\frac{x-tz}{1-t}\right)_k\right| \geq \kappa} \right) \tag{83}$$

$$\leq 2\sum_{k=1}^{d} \mathbb{E}_{x_k,t,z_k} \left( \left| \left(\frac{z-x}{1-t}\right)_k \right|^2 \mathbf{1}_{\left|\left(\frac{x-tz}{1-t}\right)_k\right| \geq \kappa} \right) + \sum_{k=1}^{d} 2\mathbb{E}_{x_k,t,z_k} \left( (u^\theta(x,t,z))_k^2 \mathbf{1}_{\left|\left(\frac{x-tz}{1-t}\right)_k\right| \geq \kappa} \right) \tag{84}$$

$$\leq 2\sum_{k=1}^{d} \mathbb{E}_{x_k,t,z_k}\left(\left|\left(\frac{z-x}{1-t}\right)_k\right|^2 \mathbf{1}_{\left|\left(\frac{x-tz}{1-t}\right)_k\right|\geq\kappa}\right) + \sum_{k=1}^{d} C_{\Phi''}\mathbb{E}_{x_k,t,z_k}\left(\left|\left(\frac{z-x}{1-t}\right)_k\right|^2 \mathbf{1}_{\left|\left(\frac{x-tz}{1-t}\right)\right|\geq\kappa}\right)$$

(85)

$$\leq (2+C_{\Phi''})\sum_{k=1}^{d}\mathbb{E}_{x_k,t,z_k}\left(\left|\left(\frac{x-z}{1-t}\right)_k\right|^2 \mathbf{1}_{\left|\left(\frac{x-tz}{1-t}\right)_k\right|\geq\kappa}\right)$$

(86)

$$\leq (2+C_{\Phi''})\sum_{k=1}^{d}\mathbb{E}_{x_k,t,z_k}\left(\left|\left(\frac{x-z}{1-t}+\frac{tz}{1-t}-\frac{tz}{1-t}\right)_k\right|^2 \mathbf{1}_{\left|\left(\frac{x-tz}{1-t}\right)_k\right|\geq\kappa}\right)$$

(87)

$$\leq (2+C_{\Phi''})\sum_{k=1}^{d} C_{\Phi''}\mathbb{E}_{x_k,t,z_k}\left(\left|\left(\frac{x-tz}{1-t}+\frac{tz-z}{1-t}\right)_k\right|^2 \mathbf{1}_{\left|\left(\frac{x-tz}{1-t}\right)_k\right|\geq\kappa}\right)$$

(88)

$$\leq (4+2C_{\Phi''})\sum_{k=1}^{d}\mathbb{E}_{x_k,t,z_k}\left(\left|\left(\frac{x-tz}{1-t}\right)_k\right|^2 \mathbf{1}_{\left|\left(\frac{x-tz}{1-t}\right)_k\right|\geq\kappa}\right)$$

$$+ (4+2C_{\Phi''})\sum_{k=1}^{d}\mathbb{E}_{x_k,t,z_k}\left(\left|\left(\frac{tz-z}{1-t}\right)_k\right|\mathbf{1}_{\left|\left(\frac{x-tz}{1-t}\right)_k\right|\geq\kappa}\right)$$

(89)

$$\leq (4+2C_{\Phi''})\left(\underbrace{\sum_{k=1}^{d}\mathbb{E}_{x_k,t_k,z_k}\left(\left|\left(\frac{x-tz}{1-t}\right)_k\right|^2 \mathbf{1}_{\left|\left(\frac{x-tz}{1-t}\right)_k\right|\geq\kappa}\right)}_{I} + \underbrace{\mathbb{E}_{x_k,t,z_k}\left(|z_k|^2 \mathbf{1}_{\left|\left(\frac{x-tz}{1-t}\right)_k\right|\geq\kappa}\right)}_{II}\right)$$

(90)

*We get the left-hand side of Equation equation 81 by expanding the definition of $\mathcal{L}(\theta)$ and $\mathcal{L}'(\theta)$ and writing the $l_2$ term as a component-wise sum. We get Equation equation 82 from Equation equation 81 by noting that $(u_t^\theta)_k = (v_t^\theta)_k$ and $(u_t^\theta(x))_k = (v_t^\theta(x))_k$ in the region where $\left|\frac{x-tz}{1-t}\right|_k \leq \kappa$. We get Equation equation 84 from Equation equation 83 by using the identity $||a-b||^2 \leq 2||a||^2 + 2||b||^2$. We get Equation equation 85 from Equation equation 84 by using Lemma D.4. We get Equation equation 89 from Equation equation 88 by using the identity $||a-b||^2 \leq 2||a||^2 + 2||b||^2$ again.*

*Now we separately obtain upper bounds for the terms I and II as follows.*

$$\sum_{k=1}^{d}\mathbb{E}_{x_k,t,z_k}\left(\left|\left(\frac{x-tz}{1-t}\right)_k\right|^2 \mathbf{1}_{\left|\left(\frac{x-tz}{1-t}\right)_k\right|\geq\kappa}\right) = \sum_{k=1}^{d}\mathbb{E}_{t,z_k}\underbrace{\mathbb{E}_{x_k|t,z_k}\left(\left|\left(\frac{x-tz}{1-t}\right)_k\right|^2 \mathbf{1}_{\left|\left(\frac{x-tz}{1-t}\right)_k\right|\geq\kappa}\right)}_{A}$$

(91)

*Now we evaluate A as follows*

$$\underbrace{\mathbb{E}_{x_k|t,z_k}\left(\left|\left(\frac{x-tz}{1-t}\right)_k\right|^2 \mathbf{1}_{\left|\left(\frac{x-tz}{1-t}\right)_k\right|\geq\kappa}\right)}_{I} = \mathbb{E}_{x\sim\mathcal{N}(0,1)}\left(x^2 \mathbf{1}_{x\geq\kappa}\right)$$

(92)

$$= \mathbb{E}_{x\sim\mathcal{N}(0,1)}\left(x^2|\mathbf{1}_{x\geq\kappa}\right)\cdot P(\mathbf{1}_{x\geq\kappa}).$$

(93)

$$\leq \mathbb{E}_{x\sim\mathcal{N}(0,1)}\left(x^2|\mathbf{1}_{x\geq\kappa}\right)\exp\left(-\kappa^2\right)$$

(94)

$$\leq \left(1+\frac{\phi(\kappa)}{1-\Phi(\kappa)}\right)\exp\left(-\frac{\kappa^2}{2C}\right)$$

(95)

$$\leq \left( 2 + \kappa^2 \mathcal{O} \left( \exp \left( -\frac{\kappa^2}{2C} \right) \right) \right) \tag{96}$$

$$\leq \mathcal{O} \left( \exp \left( -\frac{\kappa^2}{2C} \right) \right) \tag{97}$$

*We get the right hand side of Equation equation 91 by using the tower expectation property on the left hand side. We get the right-hand side of Equation equation 92 by using the fact that $x|z, t \sim \mathcal{M}(tz, (1-t)^2)$. Therefore, $\left( \frac{x-tz}{1-t} \right)_k \sim \mathcal{N}(0, 1)$ conditioned on $z$ and $t$. We get Equation equation 93 from Equation equation 92 by using Lemma D.5. We get Equation equation 95 from Equation equation 94 by using Lemma D.3. We get Equation equation 96 from Equation equation 95 by using the upper bound on the Mill's ratio which implies that $\frac{\phi(\kappa)}{1-\Phi(\kappa)} \leq \kappa + \frac{1}{\kappa}$. Plugging Equation equation 97 in to equation 91 we obtain*

$$\sum_{k=1}^{d} \mathbb{E}_{x_k, t, z_k} \left( \left| \left( \frac{x-tz}{1-t} \right)_k \right|^2 \mathbf{1}_{\left| \left( \frac{x-tz}{1-t} \right)_k \right| \geq \kappa} \right) = \mathcal{O} \left( \exp \left( -\frac{\kappa^2}{2C} \right) \right) \tag{98}$$

*We now evaluate (II) as follows*

$$\sum_{k=1}^{d} \mathbb{E}_{x_k, t, z_k} \left( |z|_k^2 \mathbf{1}_{\left| \left( \frac{x-tz}{1-t} \right)_k \right| \geq \kappa} \right) = \sum_{k=1}^{d} \mathbb{E}_{z_k, t} \mathbb{E}_{x_k|z_k, t} \left( |z_k|^2 \mathbf{1}_{\left| \left( \frac{x-tz}{1-t} \right)_k \right| \geq \kappa} \Big| z_k, t \right) \tag{99}$$

$$= \sum_{k=1}^{d} \mathbb{E}_{z_k, t} \underbrace{\mathbb{E}_{x_k|z_k, t} \left( |z_k|^2 \mathbf{1}_{\left| \left( \frac{x-tz}{1-t} \right)_k \right| \geq \kappa} \Big| z_k, t \right)}_{B} \tag{100}$$

*We get right hand side of Equation equation 99 by using tower property of expectation.*

*Now we evaluate $B$ as follows*

$$\mathbb{E}_{x_k|z_k, t} \left( |z_k|^2 \mathbf{1}_{\left| \left( \frac{x-tz}{1-t} \right)_k \right| \geq \kappa} \Big| z_k, t \right) = |z_k|^2 . \mathbb{E}_{x_k|z_k, t} \left( \mathbf{1}_{\left| \left( \frac{x-tz}{1-t} \right)_k \right| \geq \kappa} \Big| z_k, t \right) \tag{101}$$

$$\leq |z_k|^2 P \left( \left( \frac{x-tz}{1-t} \right)_k \geq \kappa \Big| z_k, t \right) \tag{102}$$

$$\leq |z_k|^2 \exp \left( -\frac{\kappa^2}{2C} \right) \tag{103}$$

*We get right hand side of Equation equation 101 by the fact that $z_k$ is constant since the expectation is conditioned on $z_k, t$. We get Equation equation 103 from equation 102 by using the fact that $x|z, t \sim \mathcal{M}(tz, (1-t)^2)$. Therefore, $\left( \frac{x-tz}{1-t} \right)_k \sim \mathcal{N}(0, 1)$ conditioned on $z$ and $t$. Plugging in Equation equation 103 into equation 100 we get*

$$\sum_{k=1}^{d} \mathbb{E}_{x_k, t, z_k} \left( |z|_k^2 \mathbf{1}_{\left| \left( \frac{x-tz}{1-t} \right)_k \right| \geq \kappa} \right) \leq \sum_{k=1}^{d} \mathbb{E}_{x_k, t, z_k} |z_k|^2 \exp \left( -\frac{\kappa^2}{2C} \right) \leq \mathcal{O} \left( \exp \left( -\frac{\kappa^2}{2C} \right) \right) \tag{104}$$

*Plugging Equation equation 98 and equation 104 into Equation equation 90 and setting $\kappa = \sqrt{2C \cdot \log \left( \frac{dn}{\delta} \right)}$, we have*

$$|\mathcal{L}(\theta) - \mathcal{L}'(\theta)| \leq \mathcal{O}\left(\frac{\delta}{d \cdot n}\right), \quad \forall \theta = \{\theta_t^a, \theta_t^b\} \tag{105}$$

*Now plugging Equation equation 105 into Equation equation 80 we get with probability at least $1 - 2\delta$*

$$|\mathcal{L}(\theta^a) - \mathcal{L}(\theta^b)| \leq \mathcal{O}\left((W)^{D-1} d \cdot \sqrt{\frac{\log \frac{1}{\delta}}{n}}\right) \tag{106}$$

*Finally, using the Polyak-Łojasiewicz (PL) condition for $\mathcal{L}(\theta)$, from Assumption 3.1, we have from the quadratic growth condition Karimi et al. (2020) of PL functions the following,*

$$\|\theta^a - \theta^b\|^2 \leq \mu \left|\mathcal{L}(\theta^a) - \mathcal{L}(\theta^b)\right|, \tag{107}$$

*and applying Lipschitz continuity of the velocity fields with respect to parameter $\theta$ from Assumption 3.2, we get*

$$\|v^{\theta^a}(x,t,z) - v^{\theta^b}(x,t,z)\|^2 \leq L \cdot \|\theta^a - \theta^b\|^2 \tag{108}$$

$$\leq L \cdot \mu \left|\mathcal{L}(\theta^a) - \mathcal{L}(\theta^b)\right| \tag{109}$$

$$\leq \mathcal{O}\left((W)^{D-1} d \cdot \sqrt{\frac{\log \frac{1}{\delta}}{n}}\right). \tag{110}$$

*Taking expectation with respect to $x, t, z$ on both sides we get*

$$\mathbb{E}_{x,t,z}\|v^{\theta^a}(x,t,z) - v^{\theta^b}(x,t,z)\|^2 \leq L \cdot \|\theta^a - \theta^b\|^2 \tag{111}$$

$$\leq \mathcal{O}\left((W)^{D-1} d \cdot \sqrt{\frac{\log \frac{1}{\delta}}{n}}\right). \tag{112}$$

*We then obtain the following.*

$$\mathbb{E}\|u^{\theta^a}(x,t,z) - u^{\theta^b}(x,t,z)\|^2 \leq 2\mathbb{E}\|u^{\theta^a}(x,t,z) - u^{\theta^b}(x,t,z) - v^{\theta^a}(x,t,z) + v^{\theta^b}(x,t,z) + v^{\theta^a}(x,t,z) - v^{\theta^b}(x,t,z)\|^2 \tag{113}$$

$$\leq 4\mathbb{E}\|u^{\theta^a}(x,t,z) - v^{\theta^a}(x,t,z)\|^2 + 4\mathbb{E}\|u^{\theta^b}(x,t,z) - v^{\theta^b}(x,t,z)\|^2$$
$$+ 8\mathbb{E}\|v^{\theta^a}(x,t,z) - v^{\theta^b}(x,t,z)\|^2 \tag{114}$$

$$\leq 4\mathbb{E}\|u^{\theta^a}(x,t,z) - v^{\theta^a}(x,t,z)\|^2 + 4\mathbb{E}\|u^{\theta^b}(x,t,z) - v^{\theta^b}(x,t,z)\|^2$$
$$+ \mathcal{O}\left((W)^{D-1} d \cdot \sqrt{\frac{\log \frac{1}{\delta}}{n}}\right) \tag{115}$$

*We get Equation equation 114 from equation 113 by using the identity $||a-b||^2 \leq 2||a||^2 + 2||b||^2$ Note that the quantities $4\mathbb{E}\|u^{\theta^b}(x,t,z) - v^{\theta^b}(x,t,z)\|^2$ and $4\mathbb{E}_{x \sim u_t}\|u^{\theta^b}(x,t,z) - v^{\theta^b}(x,t,z)\|^2$ in Equation equation 115 can be written as follows.*

$$\mathbb{E}||u^\theta(x,t,z,) - v^\theta(x,t,z)||^2 = \mathbb{E}\left\|u^\theta(x,t,z)\right\|^2 \cdot \mathbf{1}_{\left\|\left(\frac{z-x}{1-t}\right)_k\right\| \geq \kappa} \leq \mathbb{E}\left\|\left(\frac{z-x}{1-t}\right)\right\|^2 \mathbf{1}_{\left\|\left(\frac{z-x}{1-t}\right)_k\right\| \geq \kappa}$$

*Using the same analysis as is done from equation 82 downwards, we get $\mathbb{E}||u^\theta(x,t,z,) - v^\theta(x,t,z)||^2 \leq \mathcal{O}\left(\frac{\delta}{d \cdot n}\right)$. Plugging this result into equation 115, we get with probability at least $1 - 2.\delta$*

$$\mathbb{E}||u^{\theta^a}(x,t,z) - u^{\theta^b}(x,t,z)||^2 \leq \mathcal{O}\left((W)^{D-1}d \cdot \sqrt{\frac{\log \frac{1}{\delta}}{n}}\right) \tag{116}$$

*which is the required result.*

## B    Proof of Lemma 4.3

**Proof B.1** *Consider* online (one-pass) stochastic approximation *iterates indexed by $i = 1, 2, \ldots, n$:*

$$\theta_{i+1} = \theta_i - \eta_i g_i, \qquad g_i := \nabla_\theta \widehat{\mathcal{L}}(\theta_i; \xi_i),$$

*where $\{\mathcal{F}_i\}$ denotes the filtration generated by the algorithm up to iteration $i$, and $\xi_i$ is a fresh random seed drawn independently of $\mathcal{F}_i$ (e.g., $\xi_i = (x_i, t_i, z_i)$).Stepsizes are*

$$\eta_i = \frac{\alpha}{i + \gamma}, \qquad \alpha > 0, \ \gamma > 0, \tag{117}$$

*chosen so that*

$$\alpha\mu > 1 \quad and \quad \eta_i \leq \frac{1}{L} \ \ for \ all \ i \ \ (e.g. \ take \ \gamma \geq \alpha L). \tag{118}$$

*Define the expected suboptimality*

$$e_i := \mathbb{E}\big[\mathcal{L}(\theta_i) - \mathcal{L}^\star\big].$$

*By L-smoothness of $\mathcal{L}$ (Assumption 3.2), with $\theta_{i+1} = \theta_i - \eta_i g_i$,*

$$\mathcal{L}(\theta_{i+1}) \leq \mathcal{L}(\theta_i) + \left\langle \nabla\mathcal{L}(\theta_i), \theta_{i+1} - \theta_i \right\rangle + \frac{L}{2}\|\theta_{i+1} - \theta_i\|^2$$
$$= \mathcal{L}(\theta_i) - \eta_i \left\langle \nabla\mathcal{L}(\theta_i), g_i \right\rangle + \frac{L}{2}\eta_i^2\|g_i\|^2. \tag{119}$$

*Now take* conditional expectation *given $\mathcal{F}_i$ (and $t_i$ if $t_i$ is sampled separately). Since $\theta_i$ (hence $\nabla\mathcal{L}(\theta_i)$) is $\mathcal{F}_i$-measurable, Assumption 3.2 (conditional unbiasedness and conditional bounded variance) gives:*

$$\mathbb{E}\big[\left\langle \nabla\mathcal{L}(\theta_i), g_i \right\rangle \big| \mathcal{F}_i\big] = \left\langle \nabla\mathcal{L}(\theta_i), \mathbb{E}[g_i \mid \mathcal{F}_i] \right\rangle = \|\nabla\mathcal{L}(\theta_i)\|^2, \tag{120}$$
$$\mathbb{E}\big[\|g_i\|^2 \big| \mathcal{F}_i\big] = \|\nabla\mathcal{L}(\theta_i)\|^2 + \mathbb{E}\big[\|g_i - \nabla\mathcal{L}(\theta_i)\|^2 \big| \mathcal{F}_i\big] \leq \|\nabla\mathcal{L}(\theta_i)\|^2 + \sigma^2. \tag{121}$$

*Plugging equation 120–equation 121 into the conditional expectation of equation 119 yields*

$$\mathbb{E}[\mathcal{L}(\theta_{i+1}) \mid \mathcal{F}_i] \leq \mathcal{L}(\theta_i) - \eta_i\|\nabla\mathcal{L}(\theta_i)\|^2 + \frac{L}{2}\eta_i^2\Big(\|\nabla\mathcal{L}(\theta_i)\|^2 + \sigma^2\Big). \tag{122}$$

*If $\eta_i \leq 1/L$, then $-\eta_i + \frac{L}{2}\eta_i^2 \leq -\frac{\eta_i}{2}$, hence*

$$\mathbb{E}[\mathcal{L}(\theta_{i+1}) \mid \mathcal{F}_i] \leq \mathcal{L}(\theta_i) - \frac{\eta_i}{2}\|\nabla\mathcal{L}(\theta_i)\|^2 + \frac{L}{2}\eta_i^2\sigma^2. \tag{123}$$

*Using the PL condition (Assumption 3.1), $\|\nabla\mathcal{L}(\theta_i)\|^2 \geq 2\mu(\mathcal{L}(\theta_i) - \mathcal{L}^\star)$, so equation 123 gives*

$$\mathbb{E}[\mathcal{L}(\theta_{i+1}) - \mathcal{L}^\star \,|\, \mathcal{F}_i] \leq \big(1 - \mu\eta_i\big)\big(\mathcal{L}(\theta_i) - \mathcal{L}^\star\big) + \frac{L}{2}\eta_i^2\sigma^2. \tag{124}$$

*Finally taking total expectation (tower property) yields the recursion*

$$e_{i+1} \leq \Big(1 - \frac{\alpha\mu}{i+\gamma}\Big)e_i + \frac{\alpha^2 L\sigma^2}{2\,(i+\gamma)^2}. \tag{125}$$

*Set $p := \alpha\mu > 1$ and $b := \alpha^2 L\sigma^2/2$.*

*We now prove that any sequence $(e_i)$ satisfying*

$$e_{i+1} \leq \Big(1 - \frac{p}{i+\gamma}\Big)e_i + \frac{b}{(i+\gamma)^2}, \qquad p > 1,\ \gamma \geq 1, \tag{126}$$

*obeys, for all $i \geq 1$,*

$$e_i \leq \frac{\gamma^p e_1}{(i+\gamma)^p} + \frac{b}{p-1}\cdot\frac{1}{i+\gamma}. \tag{127}$$

*Define $v_i := (i+\gamma)^p e_i$. Multiply equation 126 by $(i+1+\gamma)^p$:*

$$\begin{aligned}
v_{i+1} &= (i+1+\gamma)^p e_{i+1} \\
&\leq (i+1+\gamma)^p\Big(1 - \frac{p}{i+\gamma}\Big)e_i + (i+1+\gamma)^p\frac{b}{(i+\gamma)^2}.
\end{aligned} \tag{128}$$

*Let $t := i + \gamma\ (\geq \gamma \geq 1)$. Then*

$$(i+1+\gamma)^p\Big(1 - \frac{p}{i+\gamma}\Big) = (t+1)^p\Big(1 - \frac{p}{t}\Big).$$

*By Bernoulli's inequality, for $u \in [0,1]$, $(1-u)^p \geq 1 - pu$. With $u = \frac{1}{t}$,*

$$1 - \frac{p}{t} \leq \Big(1 - \frac{1}{t}\Big)^p = \Big(\frac{t-1}{t}\Big)^p.$$

*Hence*

$$(t+1)^p\Big(1 - \frac{p}{t}\Big) \leq (t+1)^p\Big(\frac{t-1}{t}\Big)^p = \Big(\frac{t^2-1}{t}\Big)^p \leq t^p, \tag{129}$$

*since $t^2 - 1 \leq t^2$. Plugging equation 129 into equation 128 gives*

$$v_{i+1} \leq t^p e_i + (t+1)^p\frac{b}{t^2} = v_i + (t+1)^p\frac{b}{t^2}. \tag{130}$$

*Using the binomial expansion (or the mean-value form), for $t \geq 1$ and $p \geq 1$,*

$$(t+1)^p \leq t^p + pt^{p-1} + \frac{p(p-1)}{2}t^{p-2} \leq t^p\Big(1 + \frac{p}{t} + \frac{p(p-1)}{2t^2}\Big).$$

*Therefore*

$$\frac{(t+1)^p}{t^2} \leq t^{p-2} + p\,t^{p-3} + \frac{p(p-1)}{2}t^{p-4} \leq \Big(1 + \frac{p}{\gamma} + \frac{p(p-1)}{2\gamma^2}\Big)t^{p-2}, \tag{131}$$

*where in the last inequality we used $t \geq \gamma$ to factor out $t^{p-2}$ and bound the lower powers by constants depending only on $\gamma$ and $p$. Combining equation 130 and equation 131,*

$$v_{i+1} \leq v_i + c_{p,\gamma}\, b\, t^{p-2}, \qquad c_{p,\gamma} := 1 + \frac{p}{\gamma} + \frac{p(p-1)}{2\gamma^2}. \tag{132}$$

*Sum equation 132 from $j = 1$ to $i - 1$ (with $t_j = j + \gamma$):*

$$v_i \leq v_1 + c_{p,\gamma}\, b \sum_{j=1}^{i-1} (j+\gamma)^{p-2}.$$

*Since $p > 1$, the sum is bounded by an integral:*

$$\sum_{j=1}^{i-1} (j+\gamma)^{p-2} \leq \int_{\gamma}^{i+\gamma} t^{p-2}\, dt = \frac{(i+\gamma)^{p-1} - \gamma^{p-1}}{p-1} \leq \frac{(i+\gamma)^{p-1}}{p-1}.$$

*Therefore*

$$v_i \leq \gamma^p e_1 + \frac{c_{p,\gamma}\, b}{p-1} (i+\gamma)^{p-1}.$$

*Dividing by $(i+\gamma)^p$ yields*

$$e_i \leq \frac{\gamma^p e_1}{(i+\gamma)^p} + \frac{c_{p,\gamma}\, b}{p-1} \cdot \frac{1}{i+\gamma}. \tag{133}$$

*The $O(1/i)$ rate follows since $\alpha.\mu \geq 1$.*

*We state the optimization error bound after $n$ online (one-pass) stochastic gradient steps under the PL condition and the stochastic-gradient oracle assumptions. Concretely, by a standard martingale-concentration upgrade of the PL-SGD recursion, we have:*

$$\Pr\left( L(\theta_n) - L^\star \leq C_{\text{opt}} \frac{\log(1/\delta)}{n} \right) \geq 1 - \delta. \tag{134}$$

*On the event in equation 134, we control the deviation between the learned velocity $u_{\theta_n}$ and the reference velocity $u_{\theta_a}$ by (i) Lipschitzness of $u_\theta$ in $\theta$ and (ii) the PL inequality which relates $\|\theta_n - \theta_a\|^2$ to $L(\theta_n) - L^\star$. This yields a pointwise bound (uniform over $(x, t, z)$):*

$$\|u_{\theta_n}(x,t,z) - u_{\theta_a}(x,t,z)\|^2 \leq L\|\theta_n - \theta_a\|^2 \leq L\mu\big(L(\theta_n) - L^\star\big) \leq \mathcal{O}\left( \frac{\log(1/\delta)}{n} \right), \tag{134}$$

*Since the bound in equation 134 holds for all $(x, t, z)$ on the same event, taking expectation over the sampling of $(x, t, z)$ (or equivalently integrating along the reference path) preserves the rate:*

$$\mathbb{E}_{x,t,z}\big[\|u_{\theta_n}(x,t,z) - u_{\theta_a}(x,t,z)\|^2\big] \leq \mathcal{O}\left( \frac{\log(1/\delta)}{n} \right). \tag{135}$$

*We now relate the final learned field $u_{\theta_n}$ to the target field $u_{\theta_b}$ by inserting the intermediate comparator $u_{\theta_a}$ and applying the triangle inequality followed by Young's inequality. Importantly, this step is purely deterministic (no probability statement is attached here):*

$$\begin{aligned} \mathbb{E}_{x,t,z}\big[\|u_{\theta_n}(x,t,z) - u_{\theta_b}(x,t,z)\|^2\big] &\leq 2\,\mathbb{E}_{x,t,z}\big[\|u_{\theta_n}(x,t,z) - u_{\theta_a}(x,t,z)\|^2\big] \\ &\quad + 2\,\mathbb{E}_{x,t,z}\big[\|u_{\theta_a}(x,t,z) - u_{\theta_b}(x,t,z)\|^2\big]. \end{aligned} \tag{136}$$

*The first term on the right-hand side of equation 136 is controlled by equation 135 with failure probability at most $\delta$. For the second term, we invoke Lemma 4.2 with confidence parameter $\delta/2$, so that its corresponding*

*event holds with probability at least $1 - \delta$. Taking the intersection of these two events and applying a union bound gives an overall failure probability at most $2\delta$, hence with probability at least $1 - 2\delta$:*

$$\mathbb{E}_{x,t,z}\left[\|u_{\theta_n}(x,t,z) - u_{\theta_b}(x,t,z)\|^2\right] \leq \mathcal{O}\left(\frac{\log(1/\delta)}{n}\right)$$

$$+ \mathcal{O}\left((W)^{D-1}d\sqrt{\frac{\log(4/\delta)}{n}}\right). \tag{137}$$

*Finally, in the sample regime of interest (and as stated in the main theorem), the $\widetilde{\mathcal{O}}(1/n)$ optimization term is dominated by the $\widetilde{\mathcal{O}}(1/\sqrt{n})$ statistical term, so we can absorb it into the leading term (by increasing constants). Thus we obtain the final clean bound:*

$$\mathbb{E}_{x,t,z}\left[\|u_{\theta_n}(x,t,z) - u_{\theta_b}(x,t,z)\|^2\right] \leq \mathcal{O}\left((W)^{D-1}d\sqrt{\frac{\log(4/\delta)}{n}}\right). \tag{138}$$

## C    Final Theoretical Result

**Recall Theorem 4.1:** Under the assumptions 3.1, 3.2, 3.3 and 3.4, let the velocity field $u_t^\theta(x)$ be parameterized by a neural network with width $W$ and depth $D$,. Then,, if the number of i.i.d. training samples $n$ satisfies

$$n = \Omega\left((W)^{2D-2}d^2\epsilon^4\log\frac{2}{\delta}\right), \tag{139}$$

it follows with probability at least $1 - 2\delta$ that the learned velocity field satisfies the error guarantee

$$\mathbb{E}_{x,t,z}\left[\|u_t^\theta(x) - u_t(x)\|^2\right]dt \leq \epsilon^2. \tag{140}$$

Furthermore, the Wasserstein distance between the true distribution $\pi$ and the leaned distribution $\hat{\pi}$ is bounded as

$$W_2(\hat{\pi}_1, \pi_1) \leq \mathcal{O}(\epsilon) + \epsilon_{approx} \tag{141}$$

**Proof C.1** *Recall, from Equation equation 15, the velocity field is decomposed into three terms follows*

$$\mathbb{E}_{x,t,z}\left[\|u^\theta(x,t,z) - u_t(x)\|^2\right] \leq 4\underbrace{\mathbb{E}_{x,t,z}\left[\|u^{\theta^a}(x,t,z) - u_t(x)\|^2\right]}_{\mathcal{E}_t^{approx}} + 4\underbrace{\mathbb{E}_{x,t,z}\left[\|u^{\theta^a}(x,t,z) - u^{\theta^b}(x,t,z)\|^2\right]}_{\mathcal{E}_t^{stat}}$$

$$+ 4\underbrace{\mathbb{E}_{x,t,z}\left[\|u^\theta(x,t,z) - u^{\theta^b}(x,t,z)\|^2\right]}_{\mathcal{E}_t^{opt}}, \tag{142}$$

*Now using the Lemmas 4.1, 4.2, and 4.3, with probability at least $1 - 4\delta$ we have*

$$\mathbb{E}_{x\sim\mu_t}\left[\|u_t^\theta(x) - u_t(x)\|^2\right] \leq \epsilon_{approx} + \mathcal{O}\left((W)^{D-1}d\cdot\sqrt{\frac{\log\frac{2}{\delta}}{n}}\right) + \mathcal{O}\left((W)^{D-1}d\cdot\sqrt{\frac{\log\frac{2}{\delta}}{n}}\right) \tag{143}$$

$$= \epsilon_{approx} + \mathcal{O}\left((W)^{D-1}d\cdot\sqrt{\frac{\log\frac{2}{\delta}}{n}}\right) \tag{144}$$

Setting $n = \Omega\left(\frac{(W)^{2D-2}d^2}{\epsilon^4}\log\frac{2}{\delta}\right)$, , we have that

$$\mathbb{E}_{x,t,z}\left[\left\|u^\theta(x,t,z) - u_t(x)\right\|^2\right] \leq \epsilon^2 + \epsilon_{approx} \tag{145}$$

This completes the sample complexity results.

Finally the bound on the Wasserstein distance between the true and the learned distribution follows from Equations equation 8, equation 10 and equation 12.

## D    Intermediate Lemmas

**Lemma D.1 (Theorem 26.5 of Shalev-Shwartz & Ben-David (2014))** *Consider data $z \in Z$, the parametrized hypothesis class $h_\theta, \theta \in \Theta$, and the loss function $\ell(h_\theta, z) : \mathbb{R}^d \to \mathbb{R}$, where $|\ell(h_\theta, z)| \leq c$. We also define the following terms*

$$L_D(\theta) = \mathbb{E}\ell(h_\theta, z) \tag{146}$$

$$L_S(\theta) = \frac{1}{m}\sum_{i=1}^{n}\ell(h_\theta, z_i) \tag{147}$$

*which denote the expected and empirical loss functions respectively.*

*Then, with probability of at least $1 - \delta$, for all $h \in \mathcal{H}$,*

$$L_D(\theta) - L_S(\theta) \leq \widehat{R}(\theta) + \mathcal{O}\left(\sqrt{\frac{\ln(1/\delta)}{m}}\right). \tag{148}$$

*where $\widehat{R}(\theta) = \frac{1}{n}\mathbb{E}_\sigma\left[\max_{\theta \in \Theta''}\sum_{i=1}^n f(\theta)\sigma_i\right]$ denotes the empirical Radamacher complexity over the loss function $\ell$, hypothesis parameter set $\Theta$ and the dataset of size $n$.*

**Lemma D.2 (Extension of Massart's Lemma Bousquet et al. (2003))** *Let $\Theta''$ be a finite function class of cardinality $K$. Then, for any $\theta \in \Theta''$, we have*

$$\mathbb{E}_\sigma\left[\max_{\theta \in \Theta''}\sum_{i=1}^n f(\theta)\sigma_i\right] \leq \sqrt{\log K}\|f(\theta)\|_\infty \leq \sqrt{\log K}2\,(BW)^{D-1}B\,d\kappa \tag{149}$$

*where $\sigma_i$ are i.i.d random variables such that $\mathbb{P}(\sigma_i = 1) = \mathbb{P}(\sigma_i = -1) = \frac{1}{2}$, $D$ is the number of layers in the neural network, $W$ is the width and $B$ a constant such all parameters of the neural network are upper bounded by $B$. $\kappa$ is a constant such that inputs to the neural network are upper bounded by $\kappa$.*

**Proof D.1** *The first inequality in equation 149 follows from the Massart's Lemma.*

*We work with the $\ell_\infty$ norm. $\sigma$ is 1-Dipschitz and $\sigma(0) = 0$. Thus*

$$\|h_{\ell+1}\|_\infty = \|\sigma(W_\ell h_\ell + b_\ell)\|_\infty \leq \|W_\ell h_\ell + b_\ell\|_\infty \leq \|W_\ell\|_\infty\|h_\ell\|_\infty + \|b_\ell\|_\infty.$$

*Each entry of $b_\ell$ has magnitude $\leq B$. Hence $\|b_\ell\|_\infty \leq B$.*

*Each entry of $W_\ell$ has magnitude $\leq B$. If a matrix has $m$ columns, then $\|A\|_\infty \leq Bm$. Therefore $\|W_0\|_\infty \leq Bd$ (first layer has $d$ inputs). And for $\ell \geq 1$, $\|W_\ell\|_\infty \leq BW = \alpha$.*

*The input satisfies $\|x\|_\infty \leq \kappa$. Hence*

$$\|h_1\|_\infty \leq \|W_0\|_\infty\|h_0\|_\infty + \|b_0\|_\infty \leq (Bd)\kappa + B = B(d\kappa + 1).$$

*For $\ell \geq 1$ we have the affine recursion*

$$\|h_{\ell+1}\|_\infty \leq \alpha \|h_\ell\|_\infty + B.$$

*Unroll it for $D-1$ steps starting at $h_1$. We get*

$$\|h_D\|_\infty \leq \alpha^{D-1}\|h_1\|_\infty + B \sum_{i=0}^{D-2} \alpha^i.$$

*Insert the bound on $\|h_1\|_\infty$. This gives*

$$\|h_D\|_\infty \leq \alpha^{D-1}B(d\kappa+1) + B\sum_{i=0}^{D-2} \alpha^i.$$

*If $\alpha \neq 1$, use the geometric sum. Namely $\sum_{i=0}^{D-2} \alpha^i = \dfrac{\alpha^{D-1}-1}{\alpha-1}$. This yields the stated closed form.*

*If $\alpha > 1$, then*

$$\sum_{i=0}^{D-2} \alpha^i \leq (D-1)\alpha^{D-2}.$$

*Hence*

$$\|h_D\|_\infty \leq \alpha^{D-1}B(d\kappa+1) + B(D-1)\alpha^{D-2} = \alpha^{D-1}B\left(d\kappa+1+\frac{D-1}{\alpha}\right).$$

*If also $d\kappa \geq 1 + \dfrac{D-1}{\alpha}$, then*

$$d\kappa + 1 + \frac{D-1}{\alpha} \leq 2d\kappa.$$

*Therefore*

$$\|h_D\|_\infty \leq 2\,\alpha^{D-1}B\,d\kappa = 2\,(BW)^{D-1}B\,d\kappa.$$

*If $\alpha = 1$, the recursion is simpler. We have $\|h_{\ell+1}\|_\infty \leq \|h_\ell\|_\infty + B$. Thus $\|h_D\|_\infty \leq \|h_1\|_\infty + B(D-1)$. Insert $\|h_1\|_\infty \leq B(d\kappa+1)$. Obtain $\|h_D\|_\infty \leq B(d\kappa+D)$.*

**Lemma D.3 (Second Moment of a Symmetrically Truncated Normal)** *Let $X \sim \mathcal{N}(\mu, \sigma^2)$, and let $a > 0$. Then the second moment of $X$ conditioned on being outside the symmetric interval $[\mu - a, \mu + a]$ is given by*

$$\mathbb{E}[X^2 \mid |X - \mu| > a] = \mu^2 + \sigma^2 + \sigma a \cdot \frac{\phi\left(\frac{a}{\sigma}\right)}{1 - \Phi\left(\frac{a}{\sigma}\right)},$$

*where $\phi(z) = \frac{1}{\sqrt{2\pi}} e^{-z^2/2}$ is the standard normal probability density function (PDF), and $\Phi(z)$ is the standard normal cumulative distribution function (CDF).*

**Proof D.2** *Let $X \sim \mathcal{N}(\mu, \sigma^2)$. We aim to compute the second moment of $X$ conditioned on the event that it lies outside an interval centered at its mean*

$$\mathbb{E}[X^2 \mid |X - \mu| > a]$$

*This represents the expected squared value of $X$, given that $X$ is in the tails of the distribution (i.e., more than $a$ units away from the mean).*

*By definition, the conditional expectation is*

$$\mathbb{E}[X^2 \mid |X - \mu| > a] = \frac{\mathbb{E}[X^2 \cdot \mathbf{1}_{\{|X-\mu|>a\}}]}{\mathbb{P}(|X - \mu| > a)}$$

The numerator integrates $X^2$ over the tail regions $(-\infty, \mu - a) \cup (\mu + a, \infty)$, while the denominator is the probability mass in those same regions.

To simplify the integrals, we standardize $X$. Define the standard normal variable

$$Z = \frac{X - \mu}{\sigma} \sim \mathcal{N}(0,1) \quad \Rightarrow \quad X = \mu + \sigma Z$$

Define $\alpha = \frac{a}{\sigma}$. Then

$$|X - \mu| > a \quad \Leftrightarrow \quad |Z| > \alpha$$

Our conditional second moment becomes

$$\mathbb{E}[X^2 \mid |X - \mu| > a] = \mathbb{E}[(\mu + \sigma Z)^2 \mid |Z| > \alpha]$$

Expanding the square inside the expectation

$$(\mu + \sigma Z)^2 = \mu^2 + 2\mu\sigma Z + \sigma^2 Z^2$$

Taking the conditional expectation

$$\mathbb{E}[(\mu + \sigma Z)^2 \mid |Z| > \alpha] = \mu^2 + 2\mu\sigma\mathbb{E}[Z \mid |Z| > \alpha] + \sigma^2\mathbb{E}[Z^2 \mid |Z| > \alpha]$$

Since the standard normal distribution is symmetric and the region $|Z| > \alpha$ is also symmetric, we have

$$\mathbb{E}[Z \mid |Z| > \alpha] = 0$$

Thus, the expression simplifies to

$$\mathbb{E}[X^2 \mid |X - \mu| > a] = \mu^2 + \sigma^2\mathbb{E}[Z^2 \mid |Z| > \alpha]$$

By definition

$$\mathbb{E}[Z^2 \mid |Z| > \alpha] = \frac{\int_{|z|>\alpha} z^2\phi(z)\,dz}{\mathbb{P}(|Z| > \alpha)} = \frac{2\int_\alpha^\infty z^2\phi(z)\,dz}{2(1 - \Phi(\alpha))} = \frac{\int_\alpha^\infty z^2\phi(z)\,dz}{1 - \Phi(\alpha)}$$

Using Intergration by Parts we get,

$$\int_\alpha^\infty z^2\phi(z)\,dz = \phi(\alpha)\alpha + 1 - \Phi(\alpha)$$

Let $\phi(z) = \frac{1}{\sqrt{2\pi}}e^{-z^2/2}$ be the standard normal pdf and $\Phi$ its CDF. Define

$$I(a) = \int_a^\infty z^2\,\phi(z)\,dz.$$

*Since $\phi'(z) = -z\phi(z)$, we have $\int z\phi(z)\,dz = -\phi(z)$. Using integration by parts with $u = z$ and $dv = z\phi(z)\,dz$,*

$$I(a) = \int_a^\infty z^2\phi(z)\,dz = \left[-z\phi(z)\right]_a^\infty + \int_a^\infty \phi(z)\,dz$$
$$= a\,\phi(a) + \left(1 - \Phi(a)\right).$$

$$\boxed{\int_a^\infty z^2\phi(z)\,dz = a\,\phi(a) + 1 - \Phi(a)}.$$

*Therefore*

$$\mathbb{E}[Z^2 \mid |Z| > \alpha] = \frac{\phi(\alpha)\alpha + 1 - \Phi(\alpha)}{1 - \Phi(\alpha)} = 1 + \frac{\alpha\phi(\alpha)}{1 - \Phi(\alpha)}$$

*Substitute back into the expression for $\mathbb{E}[X^2 \mid |X - \mu| > a]$*

$$\mathbb{E}[X^2 \mid |X - \mu| > a] = \mu^2 + \sigma^2\left(1 + \frac{\alpha\phi(\alpha)}{1 - \Phi(\alpha)}\right)$$

*Recall that $\alpha = \frac{a}{\sigma}$, so the final expression becomes*

$$\mathbb{E}[X^2 \mid |X - \mu| > a] = \mu^2 + \sigma^2 + \sigma a \cdot \frac{\phi\left(\frac{a}{\sigma}\right)}{1 - \Phi\left(\frac{a}{\sigma}\right)}$$

**Lemma D.4 (Linear Growth of Finite Neural Networks)** *Let $f_\theta : \mathbb{R}^d \to \mathbb{R}$ be the output of a feedforward neural network with a finite number of layers and parameters and $\theta \in \Theta$ where $\Theta$ has a finite number of elements. Suppose that each activation function $\sigma : \mathbb{R} \to \mathbb{R}$ satisfies the growth condition*

$$|\sigma(z)| \le A + B|z|, \quad \text{for all } z \in \mathbb{R},$$

*for constants $A, B \ge 0$. Then there exists a constant $C_\Theta > 0$ such that for all $x \in \mathbb{R}^d$,*

$$|f(x)| \le C_\Theta(1 + \|x\|).$$

**Proof D.3** *We proceed by induction on the number of layers in the network.*

**Base case: One-layer network.** *Let the network be a single-layer function*

$$f(x) = \sum_{i=1}^k a_i\,\sigma(w_i^\top x + b_i),$$

*where $w_i \in \mathbb{R}^d$, $b_i \in \mathbb{R}$, and $a_i \in \mathbb{R}$. Then*

$$|f(x)| \le \sum_{i=1}^k |a_i| \cdot |\sigma(w_i^\top x + b_i)|.$$

*Using the growth condition on $\sigma$, we get*

$$|\sigma(w_i^\top x + b_i)| \le A + B|w_i^\top x + b_i| \le A + B(\|w_i\|\|x\| + |b_i|).$$

*Hence*

$$|f(x)| \le \sum_{i=1}^k |a_i|\,(A + B(\|w_i\|\|x\| + |b_i|)) = C_0 + C_1\|x\|,$$

*where $C_0, C_1$ are constants depending only on the network parameters. Therefore*

$$|f(x)| \le C(1 + \|x\|) \quad \text{with } C = \max\{C_0, C_1\}.$$

**Inductive step.** *Assume the result holds for all networks with $L$ layers, i.e., for any such network $f_L(x)$,*

$$|f_L(x)| \leq C_L(1 + \|x\|).$$

*Now consider a network with $L + 1$ layers, defined by*

$$f_{L+1}(x) = \sum_{j=1}^{k} a_j \, \sigma(f_L^{(j)}(x)),$$

*where each $f_L^{(j)}(x)$ is an output of a depth-$L$ subnetwork. By the inductive hypothesis*

$$|f_L^{(j)}(x)| \leq C_j(1 + \|x\|).$$

*Applying the activation bound*

$$|\sigma(f_L^{(j)}(x))| \leq A + B|f_L^{(j)}(x)| \leq A + BC_j(1 + \|x\|).$$

*Then*

$$|f_{L+1}(x)| \leq \sum_{j=1}^{k} |a_j| \cdot |\sigma(f_L^{(j)}(x))| \leq \sum_{j=1}^{k} |a_j|(A + BC_j(1 + \|x\|)) = C_{L+1}(1 + \|x\|),$$

*for some constant $C_{L+1} > 0$. This completes the induction.*

## Examples of Valid Activation Functions

*The condition $|\sigma(z)| \leq A + B|z|$ holds for most common activations*

- ***ReLU****: $\sigma(z) = \max(0, z) \Rightarrow |\sigma(z)| \leq |z|$*

- ***Leaky ReLU****: bounded by linear function of $|z|$*

- ***Tanh****: bounded by $1 \Rightarrow A = 1, B = 0$*

- ***Sigmoid****: bounded by 1*

**Lemma D.5** *Let $X$ be a real-valued random variable with probability density function $f_X$. Fix $k \in \mathbb{R}$ and set $A := \{X > k\}$. Assume*

$$0 \leq p := \mathbb{P}(X > k) = \int_k^\infty f_X(x) \, dx \leq 1 \quad \text{and} \quad \int_k^\infty |x| \, f_X(x) \, dx < \infty.$$

*Then*

$$\mathbb{E}\big[X \, \mathbf{1}_{\{X > k\}}\big] = \mathbb{P}(X > k) \, \mathbb{E}[X \mid X > k].$$

**Proof D.4** *By the definition of expectation via a density,*

$$\mathbb{E}\big[X \, \mathbf{1}_{\{X > k\}}\big] = \int_k^\infty x \, f_X(x) \, dx,$$

*which is finite by the hypothesis $\int_k^\infty |x| \, f_X(x) \, dx < \infty$.*

*We derive the conditional density of $X$ given $X > k$. For any Borel set $B \subset \mathbb{R}$ with $p = \mathbb{P}(X > k) > 0$,*

$$\mathbb{P}(X \in B \mid X > k) = \frac{\mathbb{P}(X \in B, \, X > k)}{\mathbb{P}(X > k)} = \frac{1}{p} \mathbb{P}\big(X \in B \cap (k, \infty)\big).$$

*Since $X$ has density $f_X$,*

$$\mathbb{P}(X \in B \mid X > k) = \frac{1}{p} \int_{B \cap (k,\infty)} f_X(x)\, dx = \int_B \left( \frac{f_X(x)}{p} \, \mathbf{1}_{(k,\infty)}(x) \right) dx.$$

*Therefore the conditional density is*

$$f_{X \mid X > k}(x) = \begin{cases} \dfrac{f_X(x)}{p}, & x > k, \\ 0, & x \le k. \end{cases}$$

*Hence,*

$$\mathbb{E}[X \mid X > k] = \int_{-\infty}^{\infty} x\, f_{X \mid X > k}(x)\, dx = \frac{1}{p} \int_k^{\infty} x\, f_X(x)\, dx.$$

*Multiplying both sides by $p$ yields*

$$\mathbb{P}(X > k)\, \mathbb{E}[X \mid X > k] = \int_k^{\infty} x\, f_X(x)\, dx = \mathbb{E}\big[ X\, \mathbf{1}_{\{X > k\}} \big].$$

