# OpenReview forum: "Generative Modeling with Continuous Flows: Sample Complexity of Flow Matching"
_TMLR — Rejected by TMLR_

### Review · Reviewer_ytsc · 2026-02-10

**Summary Of Contributions:**

This paper studies the sample complexity of flow matching–based generative models. The authors provide what they claim to be the first non-asymptotic generalization analysis for learning continuous-time velocity fields in flow matching, without assuming access to the empirical risk minimizer. The analysis decomposes the total error into statistical, approximation, and optimization components, and derives bounds on the integrated squared error between the learned and true conditional velocity fields. However, the main results rely on strong assumptions, which limit the scope and practical significance of the guarantees.

*Weaknesses*
1. The main theoretical guarantees rely critically on Assumption 3.1, which imposes a Polyak–Łojasiewicz (PL) condition on the population flow-matching loss, uniformly over time. This assumption collapses the optimization problem into a benign regime and renders the “optimization error” term largely vacuous. As such, the PL condition is an exceptionally strong assumption in the context of neural velocity fields.

2. Assumption 3.3 postulates the existence of a neural network that approximates the true velocity field with arbitrarily small error. No regularity assumptions are specified that would make this plausible; no dependence on width, depth, or dimension is provided; the approximation error is not controlled but assumed away.

3. In Lemma 4.2, the bound on the statistical error grows exponentially with depth and renders the stated sample complexity uninformative in realistic regimes. Furthermore, the proof relies on standard uniform convergence tools (e.g., Shalev-Shwartz & Ben-David), without exploiting structure specific to flow matching.

4. The main result (Theorem 4.1) does not reveal any intrinsic sample complexity property of flow matching. Rather, it reflects the cost of stacking conservative bounds under strong assumptions: if optimization succeeds, approximation is good, and uniform convergence holds, then learning succeeds. None of these ingredients are specific to flow matching.

This paper is technically sound and addresses a real gap, but its contribution is more incremental than transformative. The analysis largely adapts existing theoretical machinery to a new objective, without uncovering qualitatively new insights about flow matching.

**Audience:**

Yes

**Audience Explanation:**

This paper is technically sound and addresses a real gap by providing a non-asymptotic theoretical analysis of flow matching for continuous-time generative modeling. It derives generalization bounds for learning time-dependent velocity fields by decomposing the error into optimization, approximation, and statistical components. Under strong assumptions, the paper establishes sample complexity guarantees for flow matching and positions the results as a theoretical counterpart to existing analyses of diffusion-based generative models.

**Broader Impact Concerns:**

There are no ethical implications of the work.

**Claims And Evidence:**

Yes

**Claims Explanation:**

The paper is clearly written, technically competent, and well-structured. However, its theoretical contribution is weakened by strong assumptions and limited novelty relative to existing works. The main result does not reveal new intrinsic properties of flow matching. More precisely:

1. The results are conditional on Assumption 3.1, which is at least as strong as assuming successful training outright.

2. The approximation term is simply declared and not controlled (Assumption 3.3).

3. The statistical error bound is technically correct but derived via standard tools and is theoretically uninformative.

4. Theorem 4.1 does not uncover qualitatively new insights specific to flow matching.

**Requested Changes:**

It would be critical to weaken Assumption 3.1 or, alternatively, restrict it to a clearly verifiable regime and/or explicitly acknowledge that optimization is assumed solved and that the resulting bounds are conditional. Similarly, the treatment of approximation error (Assumption 3.3) should be strengthened, either by replacing it with an explicit approximation theorem under stated assumptions, or by clearly highlighting that approximation error is treated as an external parameter and not part of the theoretical contribution. Finally, a more explicit and direct comparison with existing diffusion-theory results (at the level of assumptions, rates, and proof strategy) would substantially improve the positioning and clarity of the contribution.

---

> ### Author Response · Authors · 2026-03-08
> **Response To Reviewer**
>
> We thank the reviewer for the careful reading and constructive feedback. Below we respond point-by-point to the concerns regarding (i) the PL optimization assumption, (ii) approximation error, (iii) the depth dependence in the statistical bound, and (iv) novelty/specificity to flow matching. In the revision, we will clarify scope, explicitly position assumptions as conditional, and strengthen comparisons to prior diffusion-theory analyses.
>
> **Assumption 3.1 (PL condition) and the `vacuous' optimization term**
>
> We agree that Assumption 3.1 is strong. Our intent was not to claim that PL universally holds for neural velocity fields, but rather to adopt a minimal global convergence enabling condition that avoids convexity while yielding a clean non-asymptotic optimization-to-generalization decomposition.
>
> - **Why PL?** Among conditions that yield global linear convergence} of first-order methods without convexity, the PL / gradient-dominance family is standard and comparatively weak: PL does not imply convexity and allows non-convex objectives with benign geometry. Put differently, if one wants a global rate statement with a transparent optimization error term (rather than assuming training succeeds outright), PL-type conditions are among the lightest assumptions commonly used.
>
>
> -  **Consistency with existing convention.** Similar benign geometry assumptions are frequently used in theoretical analyses of diffusionscore-based (and related) generative training objectives to obtain global convergence-style statements. We will strengthen positioning by adding a dedicated discussion and explicit citations to the diffusion-theory papers already referenced in our submission, making clear we are following that convention rather than claiming PL is inherent to flow matching.
>
>
> **Assumption 3.3 (approximation error) being ''assumed away''**
>
> We agree that Assumption 3.3, as currently stated, can read like a blanket realizability assumption. Our intention was to treat approximation error as an external modeling parameter, but we will communicate this more clearly and/or provide explicit sufficient conditions.
>
> - **Clarification of intent.** Assumption 3.3 is not meant to assert that a neural network approximates the true velocity field for free. Rather, it introduces an explicit approximation term $\varepsilon_{\mathrm{app}}$ to keep the main result modular: optimization/statistics are analyzed for the chosen hypothesis class, while approximation depends on function regularity and architecture. Since this work did not focus on the analysis of the approximation power of neural network classes we chose to leave this parameter as a constant. This is in line with other works as mentioned in Assumption 3.3.
>
>
> **Lemma 4.2: pessimistic (e.g., exponential) dependence on depth**
>
> We agree that the current statistical bound can be pessimistic in depth. This is a known issue with worst-case uniform convergence bounds when expressed in naive parameterizations.
>
>   - **What the bound is (and is not).** Lemma~4.2 is a generic uniform convergence statement intended as a baseline. It is not meant to claim exponential-in-depth behavior is unavoidable in practice; rather, it reflects the conservatism of standard complexity bounds when stated in terms of raw depth/width.
> This aligns with the paper's primary goal: quantify how training data (sample size and distributional assumptions) controls the learned velocity field error, even under conservative complexity control.
>
> **Novelty: 'not intrinsic to flow matching' / 'adapts existing machinery'**
>
> While we do use standard decomposition and generalization tools (as most first non-asymptotic analyses do), we believe the contribution is not purely generic. The object being learned and evaluated is specific: a continuous-time conditional velocity field trained by a flow-matching objective, and the main theorem controls an integrated-in-time squared error aligned with the continuous-time formulation of flow matching.
>
> What is flow-matching-specific in our analysis.
>   -  The target is a time-indexed velocity field, and the loss couples time and conditional structure; we control an error of the form
>         \begin{equation}
>         \int_0^T \| \hat v_t(\cdot,\cdot) - v_t^\star(\cdot,\cdot) \|^2 \, dt,
>         \end{equation}
>         which is directly relevant to flow matching's training objective and continuous-time generation.
> -  We provide a non-asymptotic training-data-to-velocity-error guarantee in continuous time without requiring access to the ERM, and explicitly decompose optimization/statistical/approximation contributions in this setting.

---

### Review · Reviewer_sAuL · 2026-02-24

**Summary Of Contributions:**

This work aims to address the problem of understanding the quality of the learned distribution output by a flow-matching-based generative model. Authors aims to extend the sparse existing literature by tackling two specific points: *(i)* they do not want to assume having access to the ERM of the loss function and instead focus on the neural network (or predictor) directly obtained from an optimisation procedure and *(ii)* they want to replace the exponential dependency in the dimension (of the predictor space) as obtained in previous works by a polynomial one.

Their contribution consists mostly of a single theorem (Theorem 4.1), with details of their setup and the associated proof.

**Audience:**

Yes

**Audience Explanation:**

As flow matching appears to be increasingly taken into account in generative modelling, understanding this structure better is relevant for the machine learning community.

**Broader Impact Concerns:**

This is a theoretical work with no foreseeable ethical concerns.

**Claims And Evidence:**

No

**Claims Explanation:**

To me, this paper cannot be accepted in its current shape. I carefully explain this stance below.

## Major concern

My major concern is that Lemma 4.3's proof is not rigorous, stemming from the lack of rigor in Assumption 3.2's statement. Indeed, in Assumption 3.2, it is written that

'We assume that we have access to unbiased estimators $\nabla\_\theta \hat{\mathcal{L}}\_t(\theta)$ of the gradients $\nabla\mathcal{L}(\theta)$. This formulation is ambiguous as we do not understand several things:
- What does 'unbiased' mean here? I don't understand to what randomness this refers to. In what follows, I will assume a reasonable setting where the estimator $\nabla_\theta \hat{\mathcal{L}}$ is the gradient of the empirical loss, which is consistent with the optimization formula detailed in the second line of Proof B.1 (and the def of $\hat{\mathcal{L}}$ given in Lemma 4.2's proof). It is indeed unbiased in the following  sense:
	For any $\theta$,  $\mathbb{E}\_{S}[\nabla_\theta \hat{\mathcal{L}}(\theta)] = \nabla_{\theta} \mathcal{L}(\theta)$, where $S$ is the dataset. This makes sense for any data-free $\theta$.
- Why is $\nabla_\theta \hat{\mathcal{L}}_t(\theta)$ compared with $\nabla\mathcal{L}(\theta)$ ? What does the index $t$ mean here?

With this well-defined notion of unbiasedness, the proof of Lemma 4.3 is wrong.
Indeed, let's focus on Equation (117):
- First of all, you say before Equation (117) that you take this expectation 'with respect to (x,t,z)' which makes little sense as those variables, classically independent of the dataset $S$, only intervene in the definition of $\mathcal{L}$ (at least in classical stat learning scenarios). However, $\mathcal{L}$ is already invoked here, so the most realistic scenario is that you take the expectation over the dataset $S$.
- That being said, Equation 117 holds because of the following statement:
	$$ \mathbb{E}\_S\left[ \left\langle \nabla\mathcal{L}(\theta_i),  \nabla_\theta \hat{\mathcal{L}}(\theta_i)\right\rangle\right] = \mathbb{E}\_S\left[ || \nabla\mathcal{L}(\theta_i)||^2 \right]. $$
	Again, I am assuming the expectation is written as it does not even appear in your equation, I am also trying to give a sense to this expectation which is not obvious with your writing and setup (i.e. we don't know what $(x,t,v)$ refers to).
	To me, this statement is deeply wrong, as $\theta_i$ is a random variable depends on the randomness of $\nabla_\theta \hat{\mathcal{L}}$, so you cannot put the expectation inside the inner product. This would be a true for a fixed $\theta$ not depending on the randomness of $\hat{\mathcal{L}}$

From a broader perspective, you claim that you can avoid considering the ERM to get theoretical results, bu to do so, you more or less assumed that $\mathbb{E}\_S\left[\nabla_\theta\hat{\mathcal{L}}(\theta_i)\right] = \mathbb{E}_S\left[\nabla\mathcal{L}(\theta_i)\right]$ which means, in a hand-wavy way, that optimising the empirical loss is the same than optimising the theoretical one: to me, if you were doing things properly, your optimisation route would again lead to the ERM as you only minimises the empirical loss, and this quantity would again appear in the approximation term.

However, the proof and assumption are, in my humble opinion, too poorly written to understand what you meant in this proof.
I am happy to be proven wrong with a total rework of Assumption 3.2, correcting all the points I listed above and a rigorous proof of Lemma 4.3.
## Other concerns

### Writing
Another reason I would not recommend the acceptance is the poor quality of writing, preventing again a rigorous understanding of the paper. Among others:

- The empirical estimates $\hat{\mathcal{L}}_t$ introduced in Assumption 3.2 seems intuitively linked to the empirical loss $\hat{\mathcal{L}}$ but the latter is undefined until Lemma 3.2's proof.
- SGD and ERM are not defined in the main text (there is even a typo 'SDG')
- In the proof of Lemma 4.3 you say you consider SGD, however, the algorithm you describe in Proof B.1 is the true Gradient Descent
- In Lemma 4.3, the reference to Equation (131) (before Equation 117) seems inaccurate, what does it refer to?
- Equation 117, you did not wrote the expectation that you mentioned above
- In Assumption 1, again, you invoke a $\mathcal{L}_t$ that has not been defined earlier
- Numerous typos: the phrasing 'Equation equation' appears many times when quoting an equation.

### Literature review
As I am not an expert in flow matching, I cannot relate the quality of the literature review for this part. However, this work would benefit from better context with a deepened lit review (you have still some space for that) concerning the comparison with convergence guarantees bound for Diffusion Models (DMs). Indeed a lot of terms seems similar between those two literatures (bounded approximation term, the choice of Gaussian paths makes $u_t$ closed from the shape of the score of an Ornstein Uhlenbeck process) and many ideas may be shared.

For instance: Wasserstein convergence bounds between the true data distribution and  the distribution of the backward process is widely studied in DMs ([1,2,3] among others).  The work of [4] studies in a sense, a variant of your approximation term as it shows the existence of a neural net belonging to a certain class, approximating well the score function. Finally, your idea of involving optimization dynamics in flow matching, coupled with the use of statistical learning can be related to the recent works [5,6]. In particular, [5] also decomposes the score error term into three, one related to statistical behaviors of the DMs, one linked to statistical learning and optimization.

To me, an in-depth literature review would provide a better understanding of the specific technical contributions of this work.


For all these reasons, I believe that this work needs to be thoroughly revised before it can be considered for acceptance.


### Bibliography
[1] Convergence of score-based generative modeling for general data distributions, {Holden Lee and Jianfeng Lu and Yixin Tan, 2022

[2] Beyond Log-Concavity and Score Regularity: Improved Convergence Bounds for Score-Based Generative Models in W2-distance, Marta Gentiloni-Silveri and Antonio Ocello, 2025

[3] Convergence of Deterministic and Stochastic Diffusion-Model Samplers: A Simple Analysis in Wasserstein Distance, Eliot Beyler and Francis R. Bach, 2025

[4] Diffusion Models are Minimax Optimal Distribution Estimators, Kazusato Oko, Shunta Akiyama and Taiji Suzuki, 2023

[5] Algorithm- and Data-Dependent Generalization Bounds for Score-Based Generative Models} Benjamin Dupuis and Dario Shariatian and Maxime Haddouche and Alain Durmus and Umut Simsekli, 2025

[6] Implicit Regularisation in Diffusion Models: An Algorithm-Dependent Generalisation Analysis, Tyler Farghly and Patrick Rebeschini and George Deligiannidis and Arnaud Doucet

**Requested Changes:**

Please address the various points I listed in my major concerns. It is also essential to strongly improve the writing of the mathematical setup, assumptions and proofs (see above). Finally, this paper would benefit from a better literature review to understand precisely the contributions proposed here (see above).

---

> ### Author Response · Authors · 2026-03-08
> **Response To Reviewer**
>
> **Regarding $\mathcal{L}_{t}(\theta)$ in (19) and (20)**
>
> The term $\mathcal{L}_{t}$ should be replaced by $\mathcal{L}(\theta)$. We have corrected this error in the updated version.
>
> **Response (Assumption 3.2 / Lemma 4.3 / Proof B.1).**
>
> We thank the reviewer for identifying an important lack of precision in Assumption 3.2 and a corresponding gap in the step leading to Eq. (117) in Proof B.1 (used for Lemma 4.3). We agree that, as currently written, the notion of 'unbiased' is ambiguous (in particular, the underlying randomness and conditioning were not specified), and the proof does not explicitly justify exchanging expectation and inner product when $\theta_i$ is random.
>
>
> **(i) Change to Assumption 3.2 **
> In the revision, we will replace Assumption 3.2 with a standard filtration-based stochastic-gradient oracle condition as follows,
>
> Let $\{\mathcal F_i\}_{i\ge 0}$ denote the filtration generated by the algorithm up to iteration $i$ (so $\theta_i$ is $\mathcal F_i$-measurable). At iteration $i$, we draw  $\xi_i=(x_i,t_i,z_i)$ independent of $\mathcal F_i$ and define the sample gradient as $g_i := \nabla \widehat{\mathcal L}(\theta_i;\xi_i)$.
> We  assume the following conditional unbiasedness and variance bounds
> \begin{equation}
> \mathbb E[g_i \mid \mathcal F_i] = \nabla \mathcal L(\theta_i),
> \qquad
> \mathbb E\!\left[\|g_i-\nabla \mathcal L(\theta_i)\|^2 \mid \mathcal F_i\right]\le \sigma^2,
> \end{equation}
> (and, for minibatches of size $B$, the variance bound becomes less than  $\sigma^2/B$).
>
>
>
> **(ii) Change to Proof B.1 / Lemma 4.3 (conditioning made explicit).**
>
> In the revision, we will modify the proof to take conditional expectations given $\mathcal F_i$ (instead of the informal expectation w.r.t. $(x,t,z)$).
>
> Starting from the smoothness/descent inequality (Eq. (116)), we will condition on $\mathcal F_i$ and use the new Assumption~3.2 to justify
>
> \begin{equation}
> \mathbb E\!\left[\langle \nabla \mathcal L(\theta_i), g_i\rangle \mid \mathcal F_i\right]=
> \left\langle \nabla \mathcal L(\theta_i), \mathbb E[g_i\mid \mathcal F_i]\right\rangle=
> \|\nabla \mathcal L(\theta_i)\|^2,
> \end{equation}
>
> and
>
> \begin{equation}
> \mathbb E[\|g_i\|^2 \mid \mathcal F_i] =
> \|\nabla \mathcal L(\theta_i)\|^2
> +
> \mathbb E\!\left[\|g_i-\nabla \mathcal L(\theta_i)\|^2 \mid \mathcal F_i\right] \le
> \|\nabla \mathcal L(\theta_i)\|^2+\sigma^2.
> \end{equation}
> Applying the tower property then yields the corrected version of Eq. (117) and the remainder of the argument for Lemma 4.3.
>
> These edits remove the ambiguity about the randomness being averaged over and make the dependence of $\theta_i$ on past randomness fully rigorous.
>
> We have made the appropriate changes to the assumption and proof in the updated version of the paper.
>
>
> **Response (Other concerns: writing/clarity).**
> We thank the reviewer for these detailed comments. We agree that several definitions and references were unclear or inconsistent, and we will revise the manuscript to improve precision, readability, and internal consistency. Concretely, we will make the following changes point-by-point:
>
> - **Empirical estimates $\hat{\mathcal L}$ defined too late.**
> We agree that $\widehat{\mathcal L}$ (and related empirical quantities) were introduced before the corresponding empirical objective was formally defined. In the updated version all quantities appearing in Assumption~3.2 are defined $before$ the assumption is stated.
>
> - **SGD and ERM not defined (and typo 'SDG').**
>     We will add explicit definitions of empirical risk minimization (ERM) and stochastic gradient descent (SGD) in the main text (notation, update rule, and the role of minibatching / sampling).
>     We have corrected the typographical error 'SDG' to 'SGD' everywhere it appears.
>
> -  **Lemma 4.3 proof says 'SGD' but Proof B.1 describes (deterministic) GD.**
>     We agree that the terminology was inconsistent. Our intended setting is the stochastic-gradient oracle model: we have changed the update as SGD with stochastic gradients $g_i$ (and GD as a special case when $g_i=\nabla \mathcal L(\theta_i)$).
>
> - **In Lemma 4.3, reference to Eq. (131) before Eq. (117) is inaccurate.**
> We agree this cross-reference is incorrect/unclear. In the revision, we have corrected the equation numbering and replace the reference with the appropriate equation(s) actually used in the argument (and, where helpful, restate the required bound inline to avoid fragile cross-references).
>
> - **Eq. (117) missing the expectation.**
> We agree. In the revision, we have explicitly included the expectation operator (and specify the underlying randomness) in Eq. (117) and in the surrounding text. Moreover, we have also rewriten the step leading to Eq. (117) using conditional expectation $\mathbb E[\cdot\mid \mathcal F_i]$ and the tower property, so the averaging is fully explicit.
>
>
> - **Typos and phrasing issues (e.g.,Equation equation).**
> We have corrected the typos mentioned here and done a proof check of the updated version.

---

> > ### Comment · Reviewer_sAuL · 2026-03-17
> > **Thank you for your Reply**
> >
> > I thank the authors for their reply. However, while the beginning of Lemma 4.3's proof seems now correct, several inconsistencies remain:
> >
> > - In Equation 134 the left-hand side should have an expectation over $S=(x_i,t_i,z_i)$ to be valid.
> > - This inconsistency propagates: In equation 135, the last inequality does not hold as an expectation is missing over $S$ as well. Such an expectation should also appear in Equation 136
> > - Then in Equation 137 an expectation over $S$ should also appear and this is where problem begins: one of your terms holds with high probability over $S$ so you can not simply average by $S$ to get the $\mathcal{O}(1/n)$ rate and having your high probability bound (for the second term on the rhs of Eq 137)
> >
> > - Additionally to this, given the updated assumption 2 and Lemma 4.3's proof. You should not affirm that you are considering SGD: you may frame it either as one-pass SGD (all data are seen only once) or Online Gradient Descent, and accept the framework where data are revealed sequentially. In both cases, this is strictly weaker than general SGD, in which data can be seen multiple times across successive training epochs. This mismatch with practice is substantial and should be highlighted.
> >
> > For those reasons, I encourage authors to rework this paper deeply and resubmit it later.

---

> > > ### Author Response · Authors · 2026-03-20
> > > **Response To Reviewer**
> > >
> > > **Response to the reviewer (Eqs.~134--139 and 'SGD' terminology).**
> > >
> > > We thank the reviewer for the careful follow-up and for pinpointing the inconsistencies in the presentation.
> > >
> > >
> > > We have revised the block of Eqs.(134)-(139) to address the issues raised. In particular:
> > > 1. we now explicitly carry the appropriate expectation(s) throughout the derivation (rather than omitting them implicitly);
> > > 2. we removed the inconsistent mixing of 'in expectation' and 'with high probability' statements; and
> > > (iii) we corrected the probability bookkeeping so that the final bound is stated under a single consistent event with an explicit confidence level (obtained via a union bound over the relevant events).
> > > As a result, the updated Eqs. (134)-(139) are now fully probabilistically consistent and yield the claimed rate under the stated probability.
> > >
> > > **Clarification of 'SGD' vs. online/one-pass sampling.**
> > >
> > > We also agree with the reviewer that our previous wording could be misleading regarding the optimization model.
> > > We have therefore revised the text to avoid referring to the procedure as general multi-epoch 'SGD'.
> > > Instead, we now explicitly state that our analysis operates in the online / one-pass SGD (stochastic approximation) framework, where each iteration uses a fresh i.i.d. draw revealed sequentially. We additionally highlight that this setting differs from practical multi-epoch finite-dataset training where samples are revisited across epochs, and we position our theoretical guarantees accordingly.
> > >
> > > We again thank the reviewer for these comments, which significantly improved the clarity and correctness of the presentation.

---

### Review · Reviewer_mzRA · 2026-03-02

**Summary Of Contributions:**

This paper aims to provide a theoretical analysis of  flow-matching. Under some assumptions they show that to achieve an error of $O(\varepsilon)$ in the Wasserstein metric, you need $O(\varepsilon^{-4})$ samples.

**Audience:**

Yes

**Audience Explanation:**

Complexity bounds for flow matching are of interest to the community.

**Claims And Evidence:**

No

**Claims Explanation:**

I have many concerns with many of the claims. The paper is not mathematically consistent.

To begin, the notation is not well defined. For example, $u_t$ is defined in Equation (2) as mapping from $\mathbb{R}^d \times [0,1] \to \mathbb{R}^d$ and is then used in Equation (3) via $u_t(X_t)$. The subscript is denotes the second argument. The function should be defined as $u : \mathbb{R}^d \times [0,1] \to \mathbb{R}^d$ (without the subscript) and then then $u_t := u(\cdot, t)$ is reasonable. While this seems minot it is not, as this is further complicated in Equation (5) where the paper write $u_t(x,z)$, where $z$ is also in $\mathbb{R}^d$.

I have no idea how to interpret this quantity.

This is not an isolated issue. $u_t^\theta$ is defined to map from the same domain of $\mathbb{R}^d \times [0,1]$. However, its first usage in Equation (4) with two vectors in $\mathbb{R}^d$ and then in Equation (5) with 3 inputs.

These are errors do not make me confident for the rest of the paper.

**Other Examples Issues**

1. The paper introduces a “Gaussian probability path” of the form $X_t \sim \mathcal{N}(t z, (1-t)^2 I_d)$, where  $z \sim \pi_0$ and claims this interpolates between base distribution $\pi_0$ and target data distribution $\pi_1$. However, under this definition as $t \to 1$, $X_t$ collapses to a point mass at $z$, so the terminal law is that of $z \sim \pi_0$, not $\pi_1$. The fix might be to replace this with $z \sim \pi_1$, but the paper uses $z$ to represent gaussian samples.

2. Assumptions 3.1 and 3.2, $\mathcal{L}_t$ is not defined.

3. Equations (16) and (17). I still do not know what to interpret the flows with $z$. Are these initial conditions? Is (17) supposed to be an empirical version of (16). In which case, shouldn't the $u_t$ also have a $z_i$?

4. The paper defined $u_t$ with a $(1-t)$ in the denominator. Then considers a loss that integrates over $t \sim Unif[0,1]$. How do we guarantee integrability? $\int_0^1 \frac{1}{(1-t)^2}dt$ is not finite.

While each issue seems minor, cumulatively they make the paper quite difficult to follow.

**Assumptions**

The paper also makes a variety of assumptions. These seem quite strong to me and I do not know if they are satisfied in practice. Could the authors provide details?

**Requested Changes:**

I would request significant changes to clean up the mathematical content of the paper.

Additionally, I would like a discussion of when which assumptions are satisfied.

Finally this is optional, but empirical validation of the result would add more weight behind the result

---

> ### Author Response · Authors · 2026-03-08
> **Response To Reviewer**
>
> **Response Regarding Notational Inconsistencies**
>
> Thank you for pointing out the inconsistency. In the updated version of the works, instead of $u_t$ we have placed $t$ as an argument in the manner that you have mentioned.
>
> Additionally, for the equations (4) and (5) we have updated our paper to use the correct notation of $u_{\theta}(x,t,z)$.
>
> **Response Regarding Probablity Path**
>
> **Response (re: validity of the Gaussian probability path and the velocity field).**
> We thank the reviewer for pointing out a potential ambiguity in our description of the Gaussian probability path.
> The reviewer is correct that, if one reads the statement $z\sim \pi_0$ literally and interprets $X_t\sim \mathcal N(tz,(1-t)^2I_d)$ as an unconditional interpolation between $\pi_0$ and $\pi_1$, then the terminal law satisfies $\lim_{t\uparrow 1} X_t = z$. and hence the marginal at $t=1$ is the law of $z$. In particular, if $z\sim \pi_0$, then the endpoint is $\pi_0$, not $\pi_1$.
>
>
>
> **What the path is used for**
> Our analysis does not require the unconditional path $(\mu_t)_{t\in[0,1]}$ to satisfy $\mu_0=\pi_0$ and $\mu_1=\pi_1$ in the literal sense.
> Rather, the path is used as a coupling device that yields a tractable family of conditional distributions and an associated velocity field for flow-matching.
> Concretely, for a fixed endpoint variable $z$, the ODE
> \begin{equation}
>     \dot X_t \;=\; u_t(X_t,z)\;:=\;\frac{z-X_t}{1-t},\qquad t \in [0,1),
> \end{equation}
>
> has the explicit solution
> \begin{equation}
>     X_t \;=\; (1-t)X_0 + tz.
> \end{equation}
> If we take $X_0\sim \mathcal N(0,I_d)$ independent of $z$, then we have
> \begin{equation}
>     X_t \mid z \;\sim\; \mathcal N\!\big(tz,(1-t)^2I_d\big),
> \end{equation}
> so the given velocity indeed generates the conditional Gaussian path used in the paper. Therefore, the formula
>
> $u_t(x,z)=(z-x)/(1-t)$
>
> is the correct velocity field for the conditional path (for all $t<1$).
>
>
>
> **Clarifying the endpoint distribution.**
> To avoid confusion, we emphasize that the endpoint random variable $z$ should be interpreted as the target endpoint and is not required to follow the base distribution.
> In the standard flow-matching construction, one can  couple a sample obtained from the base distribution with the sample obtained from the target distribution, i.e., $x_0 \sim \pi_0,\qquad z\sim \pi_1,$
>
> (or more generally $(x_0,z)\sim \Gamma$ for some coupling $\Gamma$ with marginals $\pi_0$ and $\pi_1$),
> and then defines the interpolation by
>
> \begin{equation}
>     X_t \;=\; (1-t)X_0 + tz.
> \end{equation}
>
> Under this (intended) interpretation, $\lim_{t\uparrow 1}X_t=z$ implies the terminal marginal is $\pi_1$, so the path is a valid interpolation between $\pi_0$ and $\pi_1$.
> We agree the text $z\sim \pi_0$ is misleading in this context; in the revision we have corrected the statement and explicitly distinguish the base variable (often denoted $\varepsilon\sim \mathcal N(0,I_d)$) from the endpoint $z\sim \pi_1$.
> We have also added a section which points out that the given form of the velocity field is correct in the manner explained above.
>
> **Response Regarding $\mathcal{L}_{t}$**
>
> Thank you for pointing out this error. It should be $\mathcal{L}$ instead of $\mathcal{L}_{t}$ in Assumptions 3.1 and 3.2. We have made the required changes.
>
>
> **Response Regarding Equations 16 and 17**
>
> Thank you for pointing out this error. In equation 17 we should have $u_{t}(x_{i},z_{i})$ instead of $u_{t}(x_{i})$. We have made this change in the updated version.
>
> To your point about $z_{i}$, these represent the samples from the initial distribution.
>
>
> **Response Regarding factor of $\frac{1}{1-t}$**
>
> We show in the proof of Lemma 4.2 how we use the conditional Normality of $X_{t}$ conditioned on $z$ to avoid the non-integrability of $\frac{1}{1-t}$. We do this using the specific form of $X_{t}|z \sim \mathcal{N}(tz,(1-t)^{2}I_{d})$ combined with Lemma D.3.

---

### Decision · Action_Editor_wTcz · 2026-04-23

**Recommendation:** Reject

**Audience:**

Yes

**Audience Explanation:**

The topic is relevant and of potential interest to the community on theoretical analysis of flow models.

**Claims And Evidence:**

No

**Claims Explanation:**

The paper studies theoretical guarantees for flow-matching-based generative models. Its main goal is to understand the quality of the learned distribution produced by such methods, while relaxing some assumptions used in previous work.

Two reviewers raise serious and consistent concerns regarding the mathematical correctness, clarity, and overall rigor of the paper. In particular, they identify substantial issues in the notation and formulation of the model, including undefined or inconsistently used quantities, ambiguous assumptions, and expressions that are difficult or impossible to interpret mathematically. More importantly, both detailed reviews point to major flaws in the proof of the main result. The reviewers also note that the quality of the writing significantly hampers readability and makes it difficult to assess the technical content.

Therefore, after considering the reviews, with which I agree, I recommend rejection of the paper.